# SERIES-TO-SERIES DIFFUSION BRIDGE MODEL

## ABSTRACT

Diffusion models have risen to prominence in time series forecasting, showcasing their robust capability to model complex data distributions. However, their effectiveness in deterministic predictions is often constrained by instability arising from their inherent stochasticity. In this paper, we revisit time series diffusion models and present a comprehensive framework that encompasses most existing diffusion-based methods. Building on this theoretical foundation, we propose a novel diffusion-based time series forecasting model, the Series-to-Series Diffusion Bridge Model ($S^2DBM$), which leverages the Brownian Bridge process to reduce randomness in reverse estimations and improves accuracy by incorporating informative priors and conditions derived from historical time series data. Experimental results demonstrate that $S^2DBM$ delivers superior performance in point-to-point forecasting and competes effectively with other diffusion-based models in probabilistic forecasting.

## 1 INTRODUCTION

Diffusion models (Ho et al., 2020; Song et al., 2020) have emerged as powerful tools for time series forecasting, offering the capability to model complex data distributions. Building on their success in other domains, such as computer vision (Saharia et al., 2022; Rombach et al., 2022) and natural language processing (Reid et al., 2022; Ye et al., 2023), researchers have increasingly applied diffusion models to time series prediction. This approach has shown promise in capturing the intricate temporal dependencies and uncertainty in time series data, leading to significant advancements in forecasting accuracy and reliability (Rasul et al., 2021; Tashiro et al., 2021; Alcaraz & Strodthoff, 2022; Li et al., 2024).

However, the inherent stochasticity of diffusion models makes multivariate time series forecasting challenging. Specifically, most of these methods employ a standard forward diffusion process that gradually corrupts future time series data until it converges to a standard normal distribution. Consequently, their predictions originate from pure noise, lacking temporal structure, with historical time series data merely conditioning the reverse diffusion process and offering limited improvement. This approach often results in forecasting instability and the generation of low-fidelity samples (as shown in Figure 1). While diffusion-based methods perform adequately in probabilistic forecasting, their point-to-point prediction accuracy lags behind that of deterministic models, e.g., Autoformer (Wu et al., 2021), PatchTST (Nie et al., 2022), and DLinear (Zeng et al., 2023).

To improve the deterministic estimation performance of diffusion models on time series, we first revisit and consolidate existing non-autoregressive diffusion-based time series forecasting models under a unified framework, demonstrating that these models are fundamentally equivalent, differing primarily in their choice of parameters and network architecture. Based on this framework, we propose a novel diffusion-based time series forecasting model, Series-to-Series Diffusion Bridge Model ($S^2DBM$). $S^2DBM$ employs the diffusion bridge as its foundational architecture, which proves effective for multivariate time series forecasting. Specifically, $S^2DBM$ uses the Brownian Bridge to pin down the diffusion process at both ends, reducing the instability caused by noisy input and enabling the accurate generation of future time step features from historical time series. By adjusting the posterior variance, $S^2DBM$ behaves as a deterministic generative model without any Gaussian noise, thereby ensuring stability and precise point-to-point forecasting results.

In our experiments, we employ seven real-world datasets as benchmarks, including Weather, Influenza-like Illness (ILI), Exchange Rate (Lai et al., 2018), and Electricity Transformer Tempera-

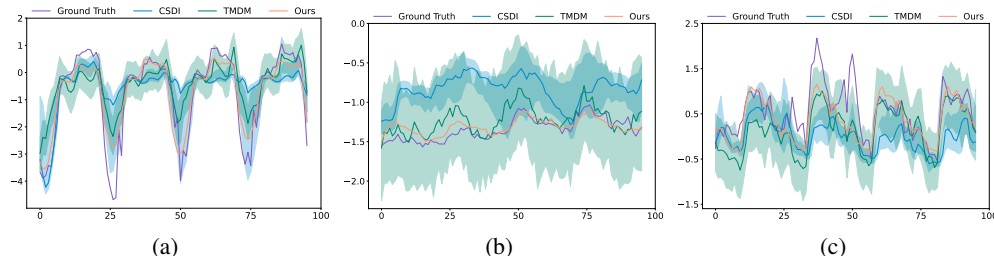

Figure 1: Examples of time series forecasting for the ETTh1 dataset. The length of forecast windows is 96. The purple line shows the ground truth. For CSDI and TMDM, median values of probabilistic forecasting are shown as the line and 5% and 95% quantiles are shown as the shade. The point-to-point forecasting results of our $S^2DBM$ are shown as the orange line.

ture datasets (ETTh1, ETTh2, ETTm1, ETTm2) (Zhou et al., 2022). We conduct experiments across various time series forecasting scenarios, covering both point-to-point and probabilistic forecasting. Through extensive testing across these scenarios, our proposed method, $S^2DBM$, demonstrates superior performance over both standard conditional diffusion-based models and a wide range of advanced time series prediction models.

Our main contributions are summarized as follows:

- In this paper, we propose a comprehensive framework for non-autoregressive time series diffusion models, into which most existing diffusion-based methods can be integrated. This framework clarifies the interrelationships between these methods and highlights practical implications for diffusion models aimed at point-to-point time series forecasting.

- Based on this framework, we introduce the Series-to-Series Diffusion Bridge Model ($S^2DBM$), which utilizes the Brownian Bridge diffusion process to reduce the randomness in reverse process of diffusion estimations. The proposed model uses linear approaches to create informative priors and conditions, thereby improving forecast accuracy by effectively using historical information for multivariate time series.

- Extensive experimental results validate the effectiveness of $S^2DBM$, which outperforms state-of-the-art time series diffusion models in point-to-point forecasting tasks. Moreover, $S^2DBM$ achieves forecasting performance on par with probabilistic models.

## 2 RELATED WORKS

**Diffusion-based Time Series Forecasting.**    Recently, a range of diffusion-based methods are proposed for time series forecasting. These methods generally adhere to the framework of the standard diffusion model, with their primary distinctions stemming from variations in the denoising network and conditional mechanisms.

TimeGrad (Rasul et al., 2021) is the pioneer of these diffusion-based methods, integrating diffusion models with an RNN-based encoder to handle historical time series. However, its reliance on autoregressive decoding can lead to error accumulation and slow inference times. To tackle this problem, CSDI (Tashiro et al., 2021) employs an entire time series as the target for diffusion and combines it with a binary mask (which denotes missing values) as conditional inputs into two transformers. This masking-based conditional mechanism enables CSDI to generate future time series data in a non-autoregressive fashion. SSSD (Alcaraz & Strodthoff, 2022) uses the same conditional mechanism as CSDI, but replaces the transformers in CSDI with a Structured State Space Model (S4) to reduce the computational complexity and is more suited to handling long-term dependencies. TMDM (Li et al., 2024) integrates transformers with a conditional diffusion process to improve probabilistic multivariate time series forecasting by effectively capturing covariate dependencies in both the forward and Reverse diffusion processes. TimeDiff (Shen & Kwok, 2023) introduces two innovative conditioning mechanisms specifically designed for time series analysis: future mixup and autoregressive initialization, which construct effective conditional embeddings. To reduce the predictive

instability arising from the stochastic nature of the diffusion models, MG-TSD (Fan et al., 2024) leverages the inherent granularity levels within the data as given targets at intermediate diffusion steps to guide the learning process of diffusion models. Most of the above diffusion-based methods emphasize their probabilistic forecasting ability; however, their performance in point-to-point forecasting is suboptimal.

**Diffusion Bridge.** Diffusion bridges (Liu et al., 2023a; Zhou et al., 2023; Li et al., 2023a) represent a specific class of diffusion models designed to simulate the trajectory of a stochastic process between predetermined initial and final states. They are regarded as conditioned diffusion models subject to particular boundary constraints. These models, stemming from classical stochastic processes like Brownian motion or Ornstein-Uhlenbeck process, have a predetermined terminal value rather than being free.

DDBMs (Zhou et al., 2023) introduce diffusion bridges, stochastically interpolating between paired distributions to provide smoother transitions and more flexible input handling compared to traditional noise-based diffusion models. Liu et al. (2023a) propose I$^2$SB, which constructs nonlinear diffusion bridges between two domains, making it suitable for tasks like image restoration. BBDM (Li et al., 2023a) models image-to-image translation as a bidirectional diffusion process using a Brownian bridge, directly learning domain translation and achieving competitive benchmark results. GOUB (Yue et al., 2023) combines the generalized OU process with Doob's h-transform to create precise diffusion mappings that transform low-quality images into high-quality ones. These diffusion bridge models excel in image restoration by using degraded images as informative priors to facilitate clean image reconstruction. Bridge-TTS (Chen et al., 2023) successfully incorporates Schrödinger Bridge diffusion models into text-to-speech (TTS) synthesis task. It leverages the latent representation obtained from text input as a prior and builds a fully tractable Schrödinger bridge between it and the ground-truth mel-spectrogram. For time series data, Park et al. (2024) introduces TimeBridge, a framework that utilizes diffusion bridges to model transitions between selected prior and data distributions. This framework supports both data- and time-dependent priors, achieving state-of-the-art performance in unconditional and conditional time series generation tasks. However, the TimeBridge uses linear spline interpolation (De Boor, 1978) to generate priors for imputation tasks, which is unsuitable for time series forecasting.

## 3 METHODOLOGY

### 3.1 PRELIMINARIES

Most diffusion-based methods for time series forecasting are designed around conditional Denoising Diffusion Probabilistic Models (DDPMs). The forward process, defined by a fixed Markov chain, progressively transforms the future time series $\boldsymbol{y} \in \mathbb{R}^{L \times d}$ into a Gaussian noise vector $\boldsymbol{y}_T$ according to a predetermined variance schedule $\{\beta_t\}_{t=1}^{T}$:

$$q\left(\boldsymbol{y}_t \mid \boldsymbol{y}_{t-1}\right) = \mathcal{N}\left(\boldsymbol{y}_t; \sqrt{1-\beta_t}\boldsymbol{y}_{t-1}, \beta_t \boldsymbol{I}\right),$$

where $L$ denotes the length of the forecast window, and $d$ represents the number of distinct features.

With the notation $\alpha_s = 1 - \beta_s$ and $\bar{\alpha}_t := \prod_{s=1}^{t} \alpha_s$, the forward process can be rewritten as:

$$\boldsymbol{y}_t = \sqrt{\bar{\alpha}_t}\boldsymbol{y}_0 + \sqrt{1-\bar{\alpha}_t}\boldsymbol{\epsilon}, \boldsymbol{\epsilon} \sim \mathcal{N}\left(\boldsymbol{0}, \boldsymbol{I}\right).$$

During inference, the model reverses the forward process by considering the following distribution:

$$p_\theta\left(\boldsymbol{y}_{0:T} \mid \boldsymbol{x}\right) = p_\theta\left(\boldsymbol{y}_T\right) \prod_{t=1}^{T} p_\theta\left(\boldsymbol{y}_{t-1} \mid \boldsymbol{y}_t, \boldsymbol{x}\right),$$

where $\boldsymbol{y}_T$ is initially sampled from a standard normal distribution $\mathcal{N}(\boldsymbol{0}, \boldsymbol{I})$, the subscripts from 0 to $T$ denote the diffusion steps. $\boldsymbol{x} \in \mathbb{R}^{H \times d}$ is the historical data, $H$ represents the length of the lookback window.

Correspondingly, the conditional reverse process at step $t$ is described by:

$$p_\theta\left(\boldsymbol{y}_{t-1} \mid \boldsymbol{y}_t, \boldsymbol{x}\right) := \mathcal{N}\left(\boldsymbol{\mu}_\theta\left(\boldsymbol{y}_t, \boldsymbol{x}, t\right), \boldsymbol{\Sigma}_\theta\left(\boldsymbol{y}_t, t\right)\right), \boldsymbol{\Sigma}_\theta\left(\boldsymbol{y}_t, t\right) = \tilde{\beta}_t = \frac{1-\bar{\alpha}_{t-1}}{1-\bar{\alpha}_t}\beta_t.$$

Table 1: Comparison between different instances of generalized conditional diffusion framework.

| Model | $\hat{\alpha}_t$ | $\hat{\beta}_t$ | $\hat{\gamma}_t$ | $\hat{\sigma}_t^2$ | Estimated Target | $E(\cdot)$ |
|---|---|---|---|---|---|---|
| CSDI (Tashiro et al., 2021) | $\sqrt{\bar{\alpha}_t}$ | $\sqrt{1-\bar{\alpha}_t}$ | 0 | $\frac{1-\bar{\alpha}_{t-1}}{1-\bar{\alpha}_t}\beta_t$ | $\epsilon$ | Transfomer in $\mu_\theta$ |
| SSSD (Alcaraz & Strodthoff, 2022) | $\sqrt{\bar{\alpha}_t}$ | $\sqrt{1-\bar{\alpha}_t}$ | 0 | $\frac{1-\bar{\alpha}_{t-1}}{1-\bar{\alpha}_t}\beta_t$ | $\epsilon$ | S4 in $\mu_\theta$ |
| TimeDiff (Shen & Kwok, 2023) | $\sqrt{\bar{\alpha}_t}$ | $\sqrt{1-\bar{\alpha}_t}$ | 0 | $\frac{1-\bar{\alpha}_{t-1}}{1-\bar{\alpha}_t}\beta_t$ | $\boldsymbol{y}_0$ | Future mixup + AR model |
| TMDM (Li et al., 2024) | $\sqrt{\bar{\alpha}_t}$ | $\sqrt{1-\bar{\alpha}_t}$ | $1-\sqrt{\bar{\alpha}_t}$ | $\frac{1-\bar{\alpha}_{t-1}}{1-\bar{\alpha}_t}\beta_t$ | $\epsilon$ | Transformer |
| Ours | $\frac{T-t}{T}$ | $\sqrt{\frac{2t(T-t)}{T^2}}$ | $\frac{t}{T}$ | $\frac{2(t-1)}{Tt}$ or 0 | $\boldsymbol{y}_0^*$ | Liner Model + Transformer in $\mu_\theta$ |

Following the formulation proposed by Saharia et al. (2022), we can parameterize $\boldsymbol{\mu}_\theta(\boldsymbol{y}_t, \boldsymbol{x}, t)$ as a neural network for either noise or data prediction. For noise prediction Tashiro et al. (2021), $\boldsymbol{\mu}_\theta$ is parameterized as:

$$\boldsymbol{\mu}_\theta(\boldsymbol{y}_t, \boldsymbol{x}, t) := \frac{1}{\sqrt{\alpha_t}}\left(\boldsymbol{y}_t - \frac{1-\alpha_t}{\sqrt{1-\bar{\alpha}_t}}\boldsymbol{\epsilon}_\theta(\boldsymbol{y}_t, \mathbf{c}, t)\right).$$

where $\boldsymbol{\epsilon}_\theta$ is a noise prediction model used to predict the noise $\boldsymbol{\epsilon}$ in the forward diffusion process, $\mathbf{c} = E(\boldsymbol{x})$ represents the condition derived from the historical data $\boldsymbol{x}$, and $E(\cdot)$ is a conditioning module. Alternatively, for data prediction (Shen & Kwok, 2023), $\boldsymbol{\mu}_\theta$ is parameterized as:

$$\boldsymbol{\mu}_\theta(\boldsymbol{y}_t, \boldsymbol{x}, t) := \frac{\sqrt{\alpha_t}(1-\bar{\alpha}_{t-1})}{1-\bar{\alpha}_t}\boldsymbol{y}_t + \frac{\sqrt{\bar{\alpha}_{t-1}}\beta_t}{1-\bar{\alpha}_t}\boldsymbol{y}_\theta(\boldsymbol{y}_t, \mathbf{c}, t),$$

where $\boldsymbol{y}_\theta$ is a data prediction model used to predict the ground truth $\boldsymbol{y}_0$.

### 3.2 REVISITING GENERALIZED DIFFUSION MODEL FOR TIME SERIES

Most existing diffusion-based time series forecasting methods emphasize their probabilistic forecasting capabilities; however, their performance in point-to-point forecasting remains suboptimal. To develop a specialized diffusion-based model tailored for point-to-point time series forecasting, a deeper understanding of existing approaches is crucial. Therefore, we revisit and consolidate current non-autoregressive diffusion-based time series forecasting models into a unified framework, demonstrating their fundamental equivalence. The primary differences among these models lie in their choice of diffusion-related coefficients and the design of network architectures.

Recognizing components in existing models, diffusion processes can be viewed in a flexible and adaptable manner. As shown in Eq. (1), the diffusion process incorporates historical data and endows the designed models with distinct properties by adjusting the coefficients $\hat{\alpha}_t$, $\hat{\beta}_t$, $\hat{\gamma}_t$, and $\hat{\sigma}_t^2$.

**Theorem 1.** *The non-autoregressive diffusion processes in time series can be formalized as follows:*

$$\boldsymbol{y}_t = \hat{\alpha}_t \boldsymbol{y}_0 + \hat{\beta}_t \boldsymbol{\epsilon} + \hat{\gamma}_t \boldsymbol{h}, \quad \boldsymbol{\epsilon} \sim \mathcal{N}(\boldsymbol{0}, \boldsymbol{I}). \tag{1}$$

*The reverse diffusion process corresponding to $\hat{\beta}_t \neq 0$ can be formulated as:*

$$p_\theta(\boldsymbol{y}_{0:T} \mid \boldsymbol{x}) := p_\theta(\boldsymbol{y}_T)\prod_{t=1}^T p_\theta(\boldsymbol{y}_{t-1} \mid \boldsymbol{y}_t, \boldsymbol{x}), \tag{2}$$

$$p_\theta(\boldsymbol{y}_{t-1} \mid \boldsymbol{y}_t, \boldsymbol{x}) := \mathcal{N}(\boldsymbol{y}_{t-1}; \mu_\theta(\boldsymbol{y}_t, \boldsymbol{h}, \mathbf{c}, t), \hat{\sigma}_t^2 \boldsymbol{I}), \tag{3}$$

*where $\hat{\alpha}_t$, $\hat{\beta}_t$, and $\hat{\gamma}_t$ are time-dependent scaling factors, these parameters are designed to ensure that $x_t$ remains pristine at $t = 0$ and undergoes maximal degradation at $t = T$. The vector $\boldsymbol{h} = F(\boldsymbol{x})$ acts as the conditional representation incorporating prior knowledge, with $F(\cdot)$ serving as the prior predictor that maps historical time series into a latent space. The initial distribution is given by $p_\theta(\boldsymbol{y}_T) = \mathcal{N}(\hat{\gamma}_T \boldsymbol{h}, \hat{\beta}_T^2 \boldsymbol{I})$. The conditioning variable $\mathbf{c} = E(\boldsymbol{x})$ guides the reverse process, where $E(\cdot)$ denotes the conditioning module. The function $\mu_\theta$ predicts the mean of $\boldsymbol{y}_{t-1}$ given inputs $\boldsymbol{y}_t$, $\boldsymbol{h}$, and $\mathbf{c}$, while $\hat{\sigma}_t^2$ represents the reverse variance schedule.*

Most existing diffusion-based time series forecasting models, including CSDI (Tashiro et al., 2021), SSSD (Alcaraz & Strodthoff, 2022), TimeDiff (Shen & Kwok, 2023), and TMDM (Li et al., 2024), can be interpreted within our proposed framework, as summarized in Table 1. The key differences lie in the choice of forward variance schedule $\hat{\gamma}_t$, the learning objectives of their denoising networks,

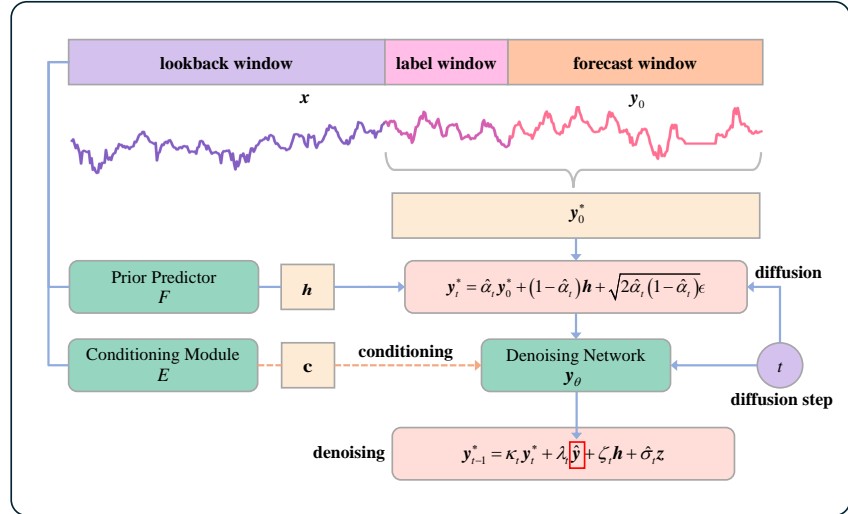

Figure 2: An illustration of the proposed $S^2$DBM

the architectures of their conditional networks $E(\cdot)$, and their respective conditioning mechanisms. Specifically, CSDI, SSSD, and TimeDiff utilize identical diffusion coefficients with $\gamma_t = 0$, aligning with the standard diffusion process. In contrast, TMDM sets $\gamma_t = 1 - \sqrt{\bar{\alpha}_t}$, introducing a distinct variance schedule. Regarding the estimation targets, CSDI, SSSD, and TMDM focus on predicting the noise component $\epsilon$, whereas TimeDiff directly estimates the data $y$. The conditioning strategies also differ notably: CSDI and SSSD employ masking with zero-padding to directly condition the denoising network, implemented via Transformer and S4 blocks, respectively. TimeDiff leverages future mixup techniques and incorporates autoregressive models, while TMDM integrates a well-designed Transformer to enhance its conditioning mechanism.

### 3.3 SERIES-TO-SERIES DIFFUSION BRIDGE MODEL

As shown in Table 1, existing diffusion-based time series forecasting methods have been extensively studied using various diffusion paradigms and conditional approaches in the formulation of Theorem 1 and achieve promising predictive ability. However, most of these methods focus on the uncertainty estimation ability and typically rely on a data-to-noise diffusion process due to current conditioning mechanisms. As a result, they are often constrained by the intrinsic stochastic nature and are limited in capturing the inherent complexity and dynamic nature of real-world time series data, leading to suboptimal performance in point-to-point forecasting. To address this gap, we propose the Series-to-Series Diffusion Bridge Model ($S^2$DBM), which uses the Brownian Bridge to pin down the diffusion process at both ends, reducing the instability caused by noisy input and enabling the accurate generation of future time step features from historical time series. By adjusting the posterior variance in Theorem 1, S2DBM behaves as a deterministic generative model without any Gaussian noise, thereby ensuring stability and precise point-to-point forecasting results.

As shown in Figure 2, $S^2$DBM employs the diffusion bridge as the foundational architecture by adjusting the coefficient schedules. The diffusion bridge pins down the diffusion process at both ends, enabling the accurate generation of future time step features from historical time series data through a data-to-data process.

**Corollary 1** (Brownian Bridge between Historical and Predicted Time Series). *Let the coefficient $\hat{\alpha}_t$, constrained to be non-negative and decrease monotonically over time $t$, satisfy the boundary conditions $\hat{\alpha}_0 = 1$ and $\hat{\alpha}_T = 0$. Additionally, define $\hat{\gamma}_t = 1 - \hat{\alpha}_t$ and $\hat{\beta}_t = \sqrt{2\hat{\alpha}_t(1 - \hat{\alpha}_t)}$ The forward process defined in Eq. (1) can be rewritten in closed form:*

$$q(\boldsymbol{y}_t \mid \boldsymbol{y}_0, \boldsymbol{h}) = \mathcal{N}(\boldsymbol{y}_t; \hat{\alpha}_t \boldsymbol{y}_0 + (1 - \hat{\alpha}_t)\boldsymbol{h}, 2\hat{\alpha}_t(1 - \hat{\alpha}_t)\boldsymbol{I}). \tag{4}$$

*Then, the reverse process transition defined in Eq. (3) turns into:*

$$p_\theta(\boldsymbol{y}_{t-1} \mid \boldsymbol{y}_t, \boldsymbol{x}) = \mathcal{N}(\boldsymbol{y}_t; \kappa_t \boldsymbol{y}_t + \lambda_t \boldsymbol{y}_\theta(\boldsymbol{y}_t, \boldsymbol{h}, \mathbf{c}, t) + \zeta_t \boldsymbol{h}, \hat{\sigma}_t^2 \boldsymbol{I}), \tag{5}$$

*here, $\kappa_t$, $\lambda_t$, and $\zeta_t$ are scaling factors defined as*

$$\kappa_t = \sqrt{\frac{2\hat{\alpha}_{t-1}(1 - \hat{\alpha}_{t-1}) - \hat{\sigma}_t^2}{2\hat{\alpha}_t(1 - \hat{\alpha}_t)}}, \quad \lambda_t = \hat{\alpha}_{t-1} - \hat{\alpha}_t\kappa_t, \quad \zeta_t = 1 - \hat{\alpha}_{t-1} - \kappa_t(1 - \hat{\alpha}_t). \quad (6)$$

Based on the Corollary 1, S$^2$DBM constructs a Brownian bridge between the initial state $y$ and the destination state $h$, eliminating the need to sample from a noisy Gaussian prior during the sampling process, allowing for the direct assignment of $y_T = h$. This approach captures more structural information about the target time series.

In the reverse process of S$^2$DBM, the diffusion process starts directly from $y_T = h$. According to Eq. (5), the mean of the reverse transition is determined by both the posterior variance $\hat{\sigma}_t^2$ and the coefficient $\hat{\alpha}_t$. Given $\hat{\alpha}_t$, the coefficients $\kappa_t$, $\lambda_t$, and $\zeta_t$ for the reverse process are analytically derived as functions of $\hat{\sigma}_t^2$. To control the contributions of $\hat{y}$, $y_t$, and $h$ to the predicted mean of $p_\theta$, following BBDM (Li et al., 2023a) and I$^3$SB Wang et al. (2024), we parameterize $\hat{\sigma}_t^2$ as follows:

$$\hat{\sigma}_t^2 = s \cdot \frac{(1 - \hat{\alpha}_{t-1})(\hat{\alpha}_{t-1} - \hat{\alpha}_t)}{1 - \hat{\alpha}_t},$$

where $s$ is a hyperparameter that scales the variance, and the selection of its numerical value is discussed in the following remark.

---

*Remark* 1 (The reverse process of S$^2$DBM). For a given trained $y_\theta$, $\hat{y} = y_\theta(y_t, h, \mathbf{c}, t)$,

- if $s = 0$, then $\hat{\sigma}_t^2 = 0$, $\kappa_t = \sqrt{\frac{\hat{\alpha}_{t-1}(1 - \hat{\alpha}_{t-1})}{\hat{\alpha}_t(1 - \hat{\alpha}_t)}}$, and the reverse process is

$$p_\theta(y_{t-1} \mid y_t, x) = \mathcal{N}(y_t; \kappa_t y_t + (\hat{\alpha}_{t-1} - \hat{\alpha}_t\kappa_t)\hat{y} + (1 - \hat{\alpha}_{t-1} - (1 - \hat{\alpha}_t)\kappa_t)h, 0).$$

In this case, the reverse process is a linear combination of $y_t$, $\hat{y}$, and $h$.

- else if $s \neq 0$, the reverse process transition is calculated according to Eq. (5) and Eq. (6). In particular, if $s = 2$, then $\hat{\sigma}_t^2 = \frac{2(1 - \hat{\alpha}_{t-1})(\hat{\alpha}_{t-1} - \hat{\alpha}_t)}{1 - \hat{\alpha}_t}$, which exhibits a form consistent with $\tilde{\beta}_t$ of DDPM; subsequently, the transition in the reverse process is

$$p_\theta(y_{t-1} \mid y_t, x) = \mathcal{N}(y_t; \frac{1 - \hat{\alpha}_{t-1}}{1 - \hat{\alpha}_t}y_t + \frac{\hat{\alpha}_{t-1} - \hat{\alpha}_t}{1 - \hat{\alpha}_t}\hat{y}, \hat{\sigma}_t^2 I).$$

In this case, the mean of $p_\theta$ depends only on $y_t$ and $\hat{y}$.

---

As a consequence of Remark 1, we discuss two instances of the reverse process in S$^2$DBM, both of which employ the same training procedure but are specifically applied to probabilistic and point-to-point forecasting, respectively.

**Example 1** (Point-to-point forecasting). *When we set $\hat{\alpha}_t = 1 - \frac{t}{T}$ and $s = 0$, the posterior variance $\hat{\sigma}_t^2$ becomes 0, making the sampling process deterministic, akin to the DDIM approach. The reverse process of* S$^2$DBM *can be rewritten as:*

$$y_{t-1} = \sqrt{\frac{(T - t + 1)(t - 1)}{(T - t)t}}y_t + \left(\frac{T - t + 1}{T} - \sqrt{\frac{(T - t)(T - t + 1)(t - 1)}{T^2 t}}\right)\hat{y}$$

$$+ \left(\frac{t - 1}{T} - \sqrt{\frac{t(T - t + 1)(t - 1)}{T^2(T - t)}}\right)h.$$

**Example 2** (Probabilistic forecasting). *When we set $\hat{\alpha}_t = 1 - \frac{t}{T}$ and $s = 1$, the posterior variance $\hat{\sigma}_t^2$ is defined as $\frac{2(t-1)}{Tt}$. Consequently, the reverse process of* S$^2$DBM *is formulated as:*

$$y_{t-1} = \left(1 - \frac{1}{t}\right)y_t + \frac{1}{t}\hat{y} + \sqrt{\frac{2(t - 1)}{Tt}}z, \quad z \sim \mathcal{N}(\mathbf{0}, I).$$

**Algorithm 1** Training of S$^2$DBM

   **Input:** dataset $\mathcal{D}$
   **repeat**
      Sample $\boldsymbol{y}*, \boldsymbol{x} \sim \mathcal{D}$ and $t \sim \mathcal{U}\,[1, T]$
      Sample $\boldsymbol{\epsilon} \sim \mathcal{N}(\boldsymbol{0}, \boldsymbol{I})$
      $\mathbf{c} = E\,(\boldsymbol{x}),\ \boldsymbol{h} = F(\boldsymbol{x})$
      $\boldsymbol{y}_t^* = \hat{\alpha}_t \boldsymbol{y}_0^* + (1 - \hat{\alpha}_t)\boldsymbol{h} + \sqrt{2\hat{\alpha}_t(1 - \hat{\alpha}_t)}\boldsymbol{\epsilon}$
      Take gradient descent step on
          $\nabla_\theta \left\| \boldsymbol{y}_0^* - \boldsymbol{y}_\theta\,(\boldsymbol{y}_t^*, \boldsymbol{h}, \mathbf{c}, t) \right\|_2^2$
   **until** converged

**Algorithm 2** Sampling of S$^2$DBM

   **Input:** $\boldsymbol{y}_T^* = \boldsymbol{h} = F(\boldsymbol{x})$, $\mathbf{c} = E\,(\boldsymbol{x})$, trained $F$, $E$ and $\boldsymbol{y}_\theta$
   **for** $t = T$ **to** 1 **do**
      Predict $\hat{\boldsymbol{y}}$ using $\boldsymbol{y}_\theta(\boldsymbol{y}_t, \boldsymbol{h}, \mathbf{c}, t)$
      $\hat{\sigma}_t^2 = s \cdot \frac{(1 - \hat{\alpha}_{t-1})(\hat{\alpha}_{t-1} - \hat{\alpha}_t)}{1 - \hat{\alpha}_t}$
      Sample $\boldsymbol{z} \sim \mathcal{N}(\boldsymbol{0}, \boldsymbol{I})$ if $t > 1$, else $\boldsymbol{z} = 0$
      $\boldsymbol{y}_{t-1}^* = \kappa_t \boldsymbol{y}_t^* + \lambda_t \hat{\boldsymbol{y}} + \zeta_t \boldsymbol{h} + \hat{\sigma}_t \boldsymbol{z}$
   **end for**
   $\boldsymbol{y}_0 \leftarrow \boldsymbol{y}_0^*$
   **return** $\boldsymbol{y}_0$

**Linear Model based Conditioning Method.** The condition $\mathbf{c}$ defined in Eq. (3) represents the useful information extracted from historical data $\boldsymbol{x}$, guiding the reverse process toward $\boldsymbol{y}_0$. Since the design of the conditioning module $E(\cdot)$ significantly impacts the predictive quality of the denoising network, it is a crucial aspect of time series diffusion models. In our S$^2$DBM model, we treat $E(\cdot)$ as independent of the denoising network, allowing $E(\boldsymbol{x})$ to preprocess historical data to provide an initial estimate of the future time series. This estimate is then used as the conditional input for the denoising network $\mu_\theta$, thereby simplifying the forecasting task.

The S$^2$DBM model captures conditional information from historical data not only through the conditioning module $E(\cdot)$, but also via the prior predictor $F(\cdot)$. In time series forecasting, the lookback and forecast windows often differ, and historical sequences cannot directly provide structurally informative priors for prediction targets as damaged images do in image restoration. Therefore, we cannot directly construct a diffusion bridge between historical time series $\boldsymbol{x}$ and future time series $\boldsymbol{y}$. Instead, we use the prior predictor $F(\cdot)$ to transform historical time series into a deterministic conditional representation $\boldsymbol{h}$, which serves as the endpoint of the diffusion process and provides guidance at the beginning of the reverse process. Both the conditional encoder network $E$ and the prior predictor $F(\cdot)$ in S$^2$DBM employ a simple one-layer linear model, chosen for its simplicity, explainability, and efficiency (Toner & Darlow, 2024).

**Label-Guided Data Estimation.** The learnable transfer probability $p_\theta(\boldsymbol{y}_{t-1} \mid \boldsymbol{y}_t, \boldsymbol{x})$ is an approximation of the posterior distribution $q(\boldsymbol{y}_{t-1} \mid \boldsymbol{y}_t, \boldsymbol{y}_0, \boldsymbol{x}) := \mathcal{N}(\boldsymbol{y}_{t-1}; \mu(\boldsymbol{y}_t, \boldsymbol{y}_0, \boldsymbol{x}), \hat{\sigma}_t^2 \boldsymbol{I})$. In our S$^2$DBM, the denoising network $\mu_\theta$ is designed to estimate the data rather than the noise, as we found that estimating the noise introduces more oscillations in the prediction results. Thus, $\mu_\theta$ can be expressed as:

$$\mu_\theta(\boldsymbol{y}_t, \boldsymbol{h}, \mathbf{c}, t) = \kappa_t \boldsymbol{y}_t + \lambda_t \boldsymbol{y}_\theta(\boldsymbol{y}_t, \boldsymbol{h}, \mathbf{c}, t) + \zeta_t \boldsymbol{h}. \tag{7}$$

In practice, we do not directly estimate the future time series $\boldsymbol{y}$. Instead, we utilize the labeling strategy employed in some transformer-based time series forecasting models, such as the Informer (Zhou et al., 2022). Specifically, we treat the terminal portion of the historical data, $\boldsymbol{x}$, as the label and integrate it with the future time series $\boldsymbol{y}$ along the time dimension, denoted as $\boldsymbol{y}^*$. Consequently, the denoising network $\mu_\theta$ is tasked not only with predicting future time steps but also with reconstructing the known sequence within the label length. This methodology enables the model to more effectively capture underlying patterns in the data. The training loss for S$^2$DBM is defined as follows:

$$\mathcal{L} = \sum_{t=1}^{T} \mathop{\mathbb{E}}_{q\left(\boldsymbol{y}_t^* \mid \boldsymbol{y}_0^*, \boldsymbol{h}\right)} \left\| \boldsymbol{y}_0^* - \boldsymbol{y}_\theta(\boldsymbol{y}_t^*, \boldsymbol{h}, \mathbf{c}, t) \right\|^2.$$

The denoising network of S$^2$DBM adopts the same architecture as CSDI but removes modules related to its original conditioning mechanism. The training and sampling procedures of S$^2$DBM are detailed in Algorithm 1 and Algorithm 2, respectively.

Table 2: Multivariate time series forecasting results in terms of MSE and MAE, lower values mean better performance. The 1st count indicates the numbers of best results.

| Methods | | Diffusion-based Methods | | | | | | Transformer-based Methods | | | | | | Linear Model | | | | | |
|---|---|---|---|---|---|---|---|---|---|---|---|---|---|---|---|---|---|---|---|
| | | Ours | | CSDI | | TMDM | | Autoformer | | Informer | | iTransformer | | NLinear | | DLinear | | RLinear | |
| Metric | | MSE | MAE | MSE | MAE | MSE | MAE | MSE | MAE | MSE | MAE | MSE | MAE | MSE | MAE | MSE | MAE | MSE | MAE |
| ETTh1 | 96 | **0.366** | **0.383** | 0.744 | 0.623 | 0.711 | 0.605 | 0.429 | 0.444 | 0.925 | 0.761 | 0.387 | 0.405 | 0.374 | 0.394 | 0.384 | 0.405 | **0.366** | 0.391 |
| | 192 | 0.405 | **0.407** | 0.952 | 0.715 | 0.922 | 0.720 | 0.440 | 0.451 | 0.995 | 0.778 | 0.441 | 0.436 | 0.408 | 0.415 | 0.443 | 0.450 | **0.403** | 0.412 |
| | 336 | 0.442 | 0.430 | 1.192 | 0.837 | 0.990 | 0.737 | 0.511 | 0.488 | 1.036 | 0.782 | 0.491 | 0.462 | 0.429 | 0.428 | 0.447 | 0.448 | **0.420** | **0.423** |
| | 720 | 0.469 | 0.478 | 1.822 | 1.005 | 1.152 | 0.836 | 0.499 | 0.501 | 1.175 | 0.858 | 0.509 | 0.494 | **0.441** | **0.454** | 0.504 | 0.515 | 0.442 | 0.456 |
| ETTh2 | 96 | 0.274 | **0.331** | 1.017 | 0.729 | 0.496 | 0.510 | 0.418 | 0.445 | 3.017 | 1.369 | 0.301 | 0.350 | 0.283 | 0.343 | 0.290 | 0.353 | **0.262** | **0.331** |
| | 192 | 0.354 | 0.388 | 3.417 | 1.356 | 0.578 | 0.535 | 0.435 | 0.439 | 6.348 | 2.105 | 0.380 | 0.399 | 0.356 | 0.385 | 0.388 | 0.422 | **0.320** | **0.374** |
| | 336 | 0.433 | 0.454 | 2.642 | 1.216 | 0.715 | 0.598 | 0.480 | 0.481 | 5.628 | 1.998 | 0.424 | 0.432 | 0.362 | 0.403 | 0.463 | 0.473 | **0.326** | **0.388** |
| | 720 | 0.592 | 0.568 | 3.396 | 1.431 | 0.758 | 0.658 | 0.478 | 0.487 | 4.110 | 1.692 | 0.430 | 0.447 | 0.398 | 0.437 | 0.733 | 0.606 | **0.425** | 0.449 |
| ETTm1 | 96 | **0.293** | **0.333** | 0.556 | 0.509 | 0.547 | 0.512 | 0.471 | 0.463 | 0.621 | 0.557 | 0.342 | 0.377 | 0.306 | 0.348 | 0.301 | 0.345 | 0.301 | 0.343 |
| | 192 | **0.333** | **0.355** | 0.608 | 0.532 | 0.689 | 0.592 | 0.592 | 0.521 | 0.723 | 0.618 | 0.383 | 0.396 | 0.349 | 0.375 | 0.336 | 0.366 | 0.341 | 0.367 |
| | 336 | **0.367** | **0.377** | 0.764 | 0.622 | 0.722 | 0.602 | 0.503 | 0.486 | 1.001 | 0.746 | 0.418 | 0.418 | 0.375 | 0.388 | 0.372 | 0.389 | 0.374 | 0.386 |
| | 720 | 0.442 | **0.422** | 1.071 | 0.792 | 1.072 | 0.785 | 0.751 | 0.582 | 0.980 | 0.747 | 0.487 | 0.457 | 0.433 | 0.422 | 0.427 | 0.423 | 0.430 | 0.418 |
| ETTm2 | 96 | **0.164** | **0.249** | 0.859 | 0.587 | 0.328 | 0.400 | 0.233 | 0.313 | 0.407 | 0.482 | 0.186 | 0.272 | 0.167 | 0.255 | 0.172 | 0.267 | **0.164** | 0.253 |
| | 192 | **0.219** | 0.292 | 0.907 | 0.614 | 0.415 | 0.423 | 0.278 | 0.336 | 0.807 | 0.706 | 0.254 | 0.314 | 0.221 | 0.293 | 0.237 | 0.314 | **0.219** | **0.290** |
| | 336 | 0.274 | 0.328 | 1.584 | 0.862 | 0.871 | 0.611 | 0.379 | 0.394 | 1.453 | 0.926 | 0.316 | 0.351 | 0.274 | 0.327 | 0.295 | 0.359 | **0.273** | **0.326** |
| | 720 | **0.361** | 0.389 | 2.692 | 1.202 | 1.101 | 0.739 | 0.584 | 0.473 | 3.930 | 1.469 | 0.414 | 0.407 | 0.369 | 0.385 | 0.427 | 0.439 | **0.366** | 0.385 |
| ILI | 24 | 2.241 | 0.983 | 3.942 | 1.293 | 4.005 | 1.183 | 3.405 | 1.290 | 5.104 | 1.544 | 2.405 | 0.987 | **2.022** | **0.925** | 2.280 | 1.061 | 2.036 | 0.969 |
| | 36 | 2.811 | 1.060 | 4.982 | 1.497 | 3.456 | 1.300 | 3.522 | 1.291 | 5.158 | 1.571 | 2.328 | 0.984 | 1.974 | 0.932 | 2.235 | 1.059 | **1.928** | **0.940** |
| | 48 | 3.024 | 1.084 | 4.164 | 1.331 | 3.059 | 1.124 | 3.478 | 1.294 | 5.101 | 1.565 | 2.330 | 0.990 | 1.979 | 0.955 | 2.298 | 1.079 | **1.880** | **0.931** |
| | 60 | 3.758 | 1.229 | 5.725 | 1.651 | 2.771 | 1.163 | 2.880 | 1.154 | 5.319 | 1.596 | 2.413 | 1.015 | 1.954 | 0.949 | 2.573 | 1.157 | 2.016 | 0.976 |
| Weather | 96 | **0.172** | **0.210** | 0.251 | 0.235 | 1.048 | 0.300 | 0.269 | 0.339 | 0.335 | 0.406 | 0.176 | 0.216 | 0.181 | 0.232 | 0.174 | 0.233 | 0.175 | 0.225 |
| | 192 | **0.213** | **0.249** | 0.330 | 0.294 | 2.246 | 0.372 | 0.338 | 0.395 | 0.693 | 0.599 | 0.225 | 0.257 | 0.225 | 0.268 | 0.218 | 0.278 | 0.217 | 0.259 |
| | 336 | **0.257** | **0.287** | 0.420 | 0.357 | 3.636 | 0.470 | 0.339 | 0.381 | 0.564 | 0.527 | 0.281 | 0.299 | 0.271 | 0.301 | 0.263 | 0.314 | 0.265 | 0.294 |
| | 720 | 0.343 | 0.353 | 0.538 | 0.423 | 0.795 | 0.541 | 0.429 | 0.433 | 1.105 | 0.771 | 0.358 | 0.350 | 0.339 | 0.349 | 0.332 | 0.374 | **0.329** | **0.339** |
| Exchange | 96 | 0.096 | 0.229 | 0.902 | 0.647 | 0.202 | 0.334 | 0.143 | 0.274 | 0.943 | 0.772 | 0.086 | 0.206 | 0.089 | 0.208 | **0.085** | 0.209 | 0.089 | 0.209 |
| | 192 | 0.196 | 0.334 | 1.084 | 0.744 | 0.371 | 0.466 | 0.266 | 0.377 | 1.244 | 0.882 | 0.181 | 0.304 | 0.181 | 0.300 | **0.162** | **0.296** | 0.191 | 0.309 |
| | 336 | 0.886 | 0.733 | 0.775 | 0.678 | 1.122 | 0.852 | 0.465 | 0.509 | 1.790 | 1.070 | 0.338 | 0.422 | 0.330 | 0.415 | 0.333 | 0.441 | 0.363 | 0.434 |
| | 720 | 2.479 | 1.179 | 1.306 | 0.879 | 1.206 | 0.792 | 1.088 | 0.812 | 2.905 | 1.406 | **0.853** | **0.696** | 0.925 | 0.722 | 0.898 | 0.725 | 0.963 | 0.731 |
| 1st Count | | 10 | 11 | 0 | 0 | 0 | 0 | 0 | 0 | 0 | 0 | 1 | 2 | 4 | 6 | 3 | 1 | 12 | 9 |

# 4 EXPERIMENTS

## 4.1 EXPERIMENTAL SETTINGS

**Datasets.** In this experiment, the time series forecasting benchmark datasets employed encompass several real-world datasets: Weather, Influenza-like Illness (ILI), Exchange-Rate (Lai et al., 2018), and four Electricity Transformer Temperature datasets (Zhou et al., 2022) (ETTh1, ETTh2, ETTm1, ETTm2). These datasets are extensively utilized for testing multivariate time-series forecasting models due to their diverse and representative nature, offering insights into the model's performance across different domains and conditions. Each dataset is normalized using the mean and standard deviation of the training part.

**Baselines.** We compared our method with several state-of-the-art and representative baseline models. These include Transformer-based methods: Autoformer (Wu et al., 2021), Informer (Zhou et al., 2022), and iTransformer (Liu et al., 2023b); linear models: DLinear, NLinear (Zeng et al., 2023), and RLinear (Li et al., 2023b); as well as diffusion-based time series prediction methods: CSDI (Tashiro et al., 2021), TMDM (Li et al., 2024), and TimeDiff (Shen & Kwok, 2023).

**Evaluation metrics.** To assess point-to-point forecasting performance, we employ mean squared error (MSE) and mean absolute error (MAE) as primary metrics to quantify discrepancies between forecasted and actual time series values. For evaluating the quality of probabilistic forecasts, we use the continuous ranked probability score (CRPS) (Matheson & Winkler, 1976) across individual time series dimensions and $\text{CRPS}_{\text{sum}}$ for the aggregate of all dimensions.

**Implementation details.** We trained our model using the ADAM optimizer, setting the initial learning rate at 0.0001 and parameters $\beta_1 = 0.9$ and $\beta_2 = 0.999$. We configured the number of time steps for the $S^2$DBM to be T=50 during the training and inference stages. The computational environment comprised a server with an NVIDIA GeForce RTX 3090 24GB GPU.

## 4.2 MAIN RESULTS

**Point-to-point forecasting.** Table 2 provides a detailed summary of the point-to-point time series forecasting results for Example 1 of our $S^2$DBM model, compared to other models. For diffusion-based methods, we evaluate results obtained from one-shot prediction. The first and second best

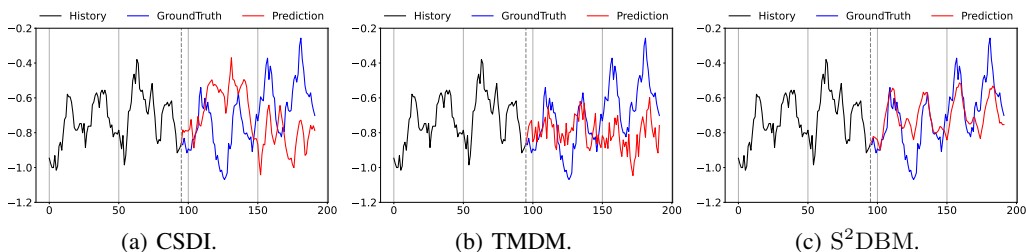

Figure 3: Visualizations on ETTh1 by CSDI, TMDM and the proposed $\mathrm{S^2DBM}$.

Table 4: Probabilistic forecasting performance comparisons on ETTh1 and ETTm1 datasets in terms of CRPS and $\mathrm{CRPS_{sum}}$. The best results are boldfaced. The prediction horizon set to 96.

| Dataset | ETTh1 | | ETTh2 | | ETTm1 | | ETTm2 | | Weather | |
|---|---|---|---|---|---|---|---|---|---|---|
| Metric | CRPS | $\mathrm{CRPS_{sum}}$ | CRPS | $\mathrm{CRPS_{sum}}$ | CRPS | $\mathrm{CRPS_{sum}}$ | CRPS | $\mathrm{CRPS_{sum}}$ | CRPS | $\mathrm{CRPS_{sum}}$ |
| CSDI | 0.512±0.107 | 2.077±0.003 | 0.579±0.096 | 2,985±0.004 | 0.428±0.106 | 2.093±0.002 | 0.490±0.104 | 2.972±0.002 | **0.190±0.026** | 1.747±0.002 |
| TMDM | 0.385±0.098 | **1.672±0.003** | 0.333±0.094 | **1.546±0.003** | 0.338±0.087 | 1.674±0.002 | **0.241±0.070** | **1.213±0.001** | 0.203±0.027 | **1.623±0.002** |
| Ours | **0.382±0.093** | 1.782±0.003 | **0.328±0.092** | 1.554±0.003 | **0.333±0.087** | **1.553±0.001** | 0.247±0.069 | 1.219±0.001 | 0.209±0.028 | 1.845±0.002 |

results are in **bold** and underlined, respectively. The smaller the value of MSE and MAE, the more accurate the prediction result is. The performance of our $\mathrm{S^2DBM}$ surpasses that of other diffusion-based methods in most cases. Compared with the Transformer-based and Linear model-based SOTA methods, our $\mathrm{S^2DBM}$ achieves the best performance on most seetings, with the 21 first and 6 second places out of 56 benchmarks in total.

Table 3 presents the Mean Squared Error (MSE) results for the diffusion-based method TimeDiff, which employed unique settings for prediction length that differ from other methods. In response, we retrain our model according to these settings and conduct the following comparisons. Experimental results indicate that our method outperforms TimeD-

Table 3: Comparison of multivariate prediction MSE between TimeDiff and $\mathrm{S^2DBM}$.

| | ETTh1 | ETTm1 | Exchange |
|---|---|---|---|
| TimeDiff | 0.407 | 0.336 | 0.018 |
| Ours | 0.397 | 0.333 | 0.018 |

iff in terms of MSE. To complement the quantitative results of diffusion-based methods, Figure 3 provides visualizations of the predictions obtained by CSDI, TMDM, and the proposed $\mathrm{S^2DBM}$ on a randomly selected test example from the ETTh1 dataset. As illustrated, while CSDI delivers accurate short-term predictions (from steps 96-110), its long-term forecasts deviate significantly from the ground truth. TMDM captures the overall trend of the future time series, but its point-wise prediction accuracy shows significant oscillations, likely influenced by the noise inherent in the diffusion process, leading to fluctuating results. In contrast, $\mathrm{S^2DBM}$ effectively captures the trend and seasonality of time series.

**Probabilistic forecasting.** Table 4 summarizes the probabilistic forecasting results for Example 2 of our $\mathrm{S^2DBM}$ model, compared with other diffusion-based models. We utilized 100 samples to approximate the probability distribution. The results show that our $\mathrm{S^2DBM}$ performs competitively against CSDI and TMDM in terms of CRPS and $\mathrm{CRPS_{sum}}$, illustrating the capabilities of our $\mathrm{S^2DBM}$ in probabilistic forecasting.

### 4.3 ABLATION STUDIES

To validate each component of our proposed $\mathrm{S^2DBM}$ model, we performed a comparative analysis of prediction results using five different models on the ETTh1 and ETTm1 datasets. The results are presented in Table 5. The notation cDDPM indicates that it employs the standard diffusion process instead of the Brownian bridge process used in $\mathrm{S^2DBM}$. The notation w/ CSDI $E$ refers to an operation that utilizes the conditioning mechanism of CSDI. Similarly, w/ CSDI $\mu_\theta$ indicates the adoption of the denoising network architecture from CSDI. Additionally, the notation label_len $= 0$ signifies that $\mathrm{S^2DBM}$ no longer reconstructs known data, focusing solely on predicting the future time

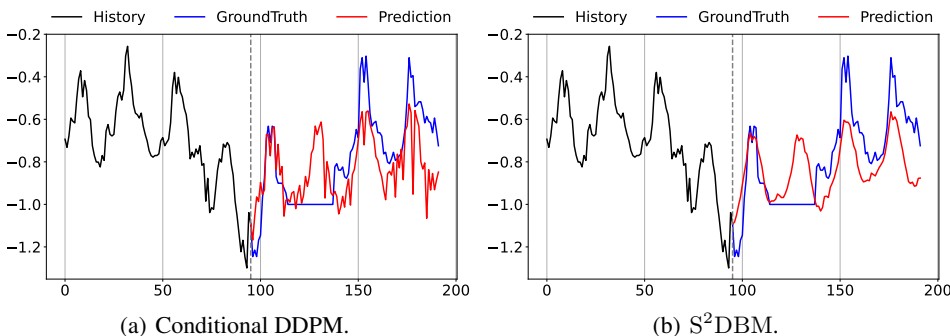

(a) Conditional DDPM.  (b) S²DBM.

Figure 4: Visualizations on ETTh1 by Conditional DDPM and the proposed S$^2$DBM.

series. When comparing our proposed model S$^2$DBM with cDDPM, we observe notable improvements in both MSE and MAE. Figure 4 visualizes the predictions obtained from both cDDPM and the proposed S$^2$DBM for a randomly selected test example from the ETTh1 dataset. As illustrated, S$^2$DBM significantly reduces oscillations in the predictions. Additionally, comparing w/ CSDI $E$ and w/ CSDI $\mu_\theta$ with S$^2$DBM demonstrates the advantages of the linear model-based conditioning method and the network architecture of S$^2$DBM. Finally, comparing S$^2$DBM with label_len $= 0$, we reveal an average reduction of 21% in MSE and 16% in MAE, indicating the contribution of the labeling strategy.

Table 5: Model ablation.We present the MSE and MAE of different variants of the S$^2$DBM model, with the prediction horizon set to 96.

| Dataset | ETTh1 | | ETTm1 | |
|---|---|---|---|---|
| Metric | MSE | MAE | MSE | MAE |
| cDDPM | 0.379 | 0.392 | 0.304 | 0.345 |
| w/ CSDI $E$ | 0.755 | 0.545 | 0.416 | 0.406 |
| w/ CSDI $\mu_\theta$ | 0.578 | 0.520 | 0.489 | 0.457 |
| label_len=0 | 0.450 | 0.461 | 0.378 | 0.396 |
| Ours | **0.366** | **0.383** | **0.293** | **0.333** |

## 5 CONCLUSION

In this paper, we revisit non-autoregressive time series diffusion models and present a comprehensive framework that integrates most existing diffusion-based methods. Building on this theoretical framework, we propose the Series-to-Series Diffusion Bridge Model (S$^2$DBM). Our S$^2$DBM utilizes the Brownian Bridge diffusion process to reduce randomness in diffusion estimations, improving forecast accuracy by effectively leveraging historical information through informative priors and conditions. Extensive experimental results demonstrate that S$^2$DBM achieves superior performance in point-to-point forecasting and performs competitively against other diffusion-based models in probabilistic forecasting.

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

## A  APPENDIX

### A.1  PROOFS OF THEOREM 1

The non-autoregressive diffusion processes in time series can be formalized as follows:

$$\boldsymbol{y}_t = \hat{\alpha}_t \boldsymbol{y}_0 + \hat{\beta}_t \boldsymbol{\epsilon}_t + \hat{\gamma}_t \boldsymbol{h}, \quad \boldsymbol{\epsilon}_t \sim \mathcal{N}(\boldsymbol{0}, \boldsymbol{I}). \tag{8}$$

Here, $\hat{\alpha}_t$, $\hat{\beta}_t$, and $\hat{\gamma}_t$ are time-dependent scaling factors, and $\boldsymbol{h} = F(\boldsymbol{x})$ serves as the conditional representation acting as prior knowledge.

Similarly, the previous state $\boldsymbol{y}_{t-1}$ can be expressed as:

$$\boldsymbol{y}_{t-1} = \hat{\alpha}_{t-1} \boldsymbol{y}_0 + \hat{\beta}_{t-1} \boldsymbol{\epsilon}_{t-1} + \hat{\gamma}_{t-1} \boldsymbol{h}, \quad \boldsymbol{\epsilon}_{t-1} \sim \mathcal{N}(\boldsymbol{0}, \boldsymbol{I}). \tag{9}$$

We are interested in the posterior distribution $q\left(\boldsymbol{y}_{t-1} \mid \boldsymbol{y}_t, \boldsymbol{y}_0, \boldsymbol{h}\right)$. According to the properties of Gaussian distributions, this posterior is also Gaussian and can be written as:

$$q\left(\boldsymbol{y}_{t-1} \mid \boldsymbol{y}_t, \boldsymbol{y}_0, \boldsymbol{h}\right) = \mathcal{N}\left(\boldsymbol{y}_{t-1}; \kappa_t \boldsymbol{y}_t + \lambda_t \boldsymbol{y}_0 + \zeta_t \boldsymbol{h}, \hat{\sigma}_t^2 \boldsymbol{I}\right), \tag{10}$$

where $\kappa_t$, $\lambda_t$, and $\zeta_t$ are coefficients to be determined, and $\hat{\sigma}_t^2$ is the variance.

By substituting Eq. (8) into the expression for $\boldsymbol{y}_{t-1}$, we obtain:

$$\begin{aligned} \boldsymbol{y}_{t-1} &= \kappa_t \boldsymbol{y}_t + \lambda_t \boldsymbol{y}_0 + \zeta_t \boldsymbol{h} + \hat{\sigma}_t \boldsymbol{\epsilon}' \\ &= \kappa_t(\hat{\alpha}_t \boldsymbol{y}_0 + \hat{\beta}_t \boldsymbol{\epsilon}_t + \hat{\gamma}_t \boldsymbol{h}) + \lambda_t \boldsymbol{y}_0 + \zeta_t \boldsymbol{h} + \hat{\sigma}_t \boldsymbol{\epsilon}' \\ &= (\kappa_t \hat{\alpha}_t + \lambda_t) \boldsymbol{y}_0 + (\kappa_t \hat{\gamma}_t + \zeta_t) \boldsymbol{h} + (\kappa_t \hat{\beta}_t \boldsymbol{\epsilon}_t + \hat{\sigma}_t \boldsymbol{\epsilon}'), \end{aligned} \tag{11}$$

where $\boldsymbol{\epsilon}' \sim \mathcal{N}(\boldsymbol{0}, \boldsymbol{I})$ is independent of $\boldsymbol{\epsilon}_t$.

Since the sum of two independent Gaussian noises is another Gaussian noise, we have:

$$\kappa_t \hat{\beta}_t \boldsymbol{\epsilon}_t + \hat{\sigma}_t \boldsymbol{\epsilon}' = \sqrt{\kappa_t^2 \hat{\beta}_t^2 + \hat{\sigma}_t^2}, \boldsymbol{\epsilon}_{t-1}, \tag{12}$$

where $\boldsymbol{\epsilon}_{t-1} \sim \mathcal{N}(\boldsymbol{0}, \boldsymbol{I})$.

Comparing this with Eq. (9), we can equate the coefficients:

$$\hat{\alpha}_{t-1} = \kappa_t \hat{\alpha}_t + \lambda_t, \quad \hat{\gamma}_{t-1} = \kappa_t \hat{\gamma}_t + \zeta_t, \quad \hat{\beta}_{t-1} = \sqrt{\kappa_t^2 \hat{\beta}_t^2 + \hat{\sigma}_t^2}. \tag{13}$$

Solving for $\kappa_t$, $\lambda_t$, and $\zeta_t$, we get:

$$\begin{aligned} \kappa_t &= \frac{\sqrt{\hat{\beta}_{t-1}^2 - \hat{\sigma}_t^2}}{\hat{\beta}_t} \\ \lambda_t &= \hat{\alpha}_{t-1} - \frac{\hat{\alpha}_t \sqrt{\hat{\beta}_{t-1}^2 - \hat{\sigma}_t^2}}{\hat{\beta}_t} = \hat{\alpha}_{t-1} - \hat{\alpha}_t \kappa_t \\ \hat{\zeta}_t &= \hat{\gamma}_{t-1} - \frac{\hat{\gamma}_t \sqrt{\hat{\beta}_{t-1}^2 - \hat{\sigma}_t^2}}{\hat{\beta}_t} = \hat{\gamma}_{t-1} - \hat{\gamma}_t \kappa_t \end{aligned} \tag{14}$$

Since $\boldsymbol{h}$ is completely determined by $\boldsymbol{x}$, the posterior distribution becomes:

$$q\left(\boldsymbol{y}_{t-1} \mid \boldsymbol{y}_t, \boldsymbol{y}_0, \boldsymbol{x}\right) = \mathcal{N}\left(\boldsymbol{y}_{t-1}; \kappa_t \boldsymbol{y}_t + \lambda_t \boldsymbol{y}_0 + \zeta_t \boldsymbol{h}, \hat{\sigma}_t^2 \boldsymbol{I}\right). \tag{15}$$

However, this posterior depends on the unknown data distribution $q(\boldsymbol{y}_0)$, making it impractical for direct use. Therefore, we introduce a learnable transition probability $p_\theta(\boldsymbol{y}_{t-1} \mid \boldsymbol{y}_t, \boldsymbol{x})$ to approximate $q\left(\boldsymbol{y}_{t-1} \mid \boldsymbol{y}_t, \boldsymbol{y}_0, \boldsymbol{x}\right)$ for all $t$. The reverse process is defined as:

$$p_\theta(\boldsymbol{y}_{0:T} \mid \boldsymbol{x}) := p_\theta(\boldsymbol{y}_T) \textstyle\prod_{t=1}^{T} p_\theta(\boldsymbol{y}_{t-1} \mid \boldsymbol{y}_t, \boldsymbol{x}), \tag{16}$$

$$p_\theta(\boldsymbol{y}_{t-1} \mid \boldsymbol{y}_t, \boldsymbol{x}) := \mathcal{N}(\boldsymbol{y}_{t-1}; \mu_\theta\left(\boldsymbol{y}_t, \boldsymbol{h}, \mathbf{c}, t\right), \hat{\sigma}_t^2 \boldsymbol{I}) \tag{17}$$

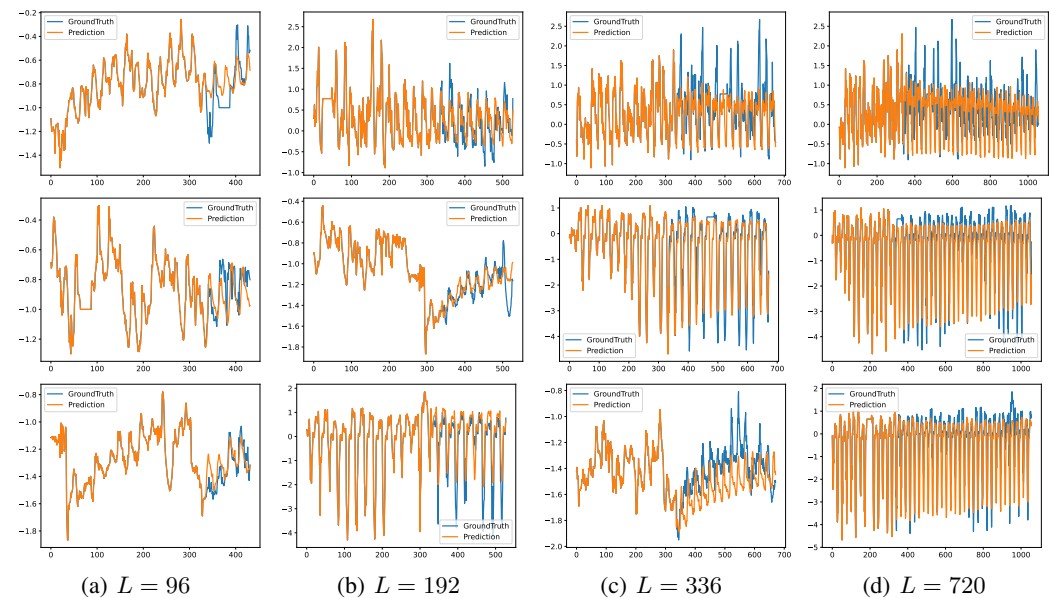

(a) $L = 96$      (b) $L = 192$      (c) $L = 336$      (d) $L = 720$

Figure 5: The predicted samples by our $\text{S}^2\text{DBM}$ model for different forecast window lengths on the ETTh1 dataset.

Here, $\mathbf{c} = \boldsymbol{E}(\boldsymbol{x})$ represents the condition guiding the reverse process, where $\boldsymbol{E}(\cdot)$ is a conditioning network taking historical data $\boldsymbol{x}$ as input, and $\theta$ includes all trainable parameters of the model. The mean $\mu_\theta$ is trained to predict $\boldsymbol{y}_{t-1}$ given $\boldsymbol{y}_t$, $\boldsymbol{h}$, and $\mathbf{c}$, with the reverse variance schedule $\hat{\sigma}_t^2$ fixed.

When we use $\boldsymbol{y}_\theta$ as the data prediction model to estimate the ground truth $\boldsymbol{y}_0$, the mean $\mu_\theta$ can be expressed as:

$$\mu_\theta(\boldsymbol{y}_t, \boldsymbol{h}, \mathbf{c}, t) = \kappa_t \boldsymbol{y}_t + \lambda_t \boldsymbol{y}_\theta(\boldsymbol{y}_t, \boldsymbol{h}, \mathbf{c}, t) + \zeta_t \boldsymbol{h}. \tag{18}$$

In this formulation, $\boldsymbol{y}_\theta(\boldsymbol{y}_t, \boldsymbol{h}, \mathbf{c}, t)$ is a neural network that predicts $\boldsymbol{y}_0$ from $\boldsymbol{y}_t$, conditioned on $\boldsymbol{h}$, $\mathbf{c}$, and time $t$.

### A.2 MORE FORECASTING RESULTS VISUALIZATION

To enhance the comprehensive understanding of our forecasting methods, we present additional visualizations of our predictive results in the following sections. These supplemental images delve deeper into the performance variations of our models under different conditions. By exploring these extra results, readers can obtain a more detailed appreciation of the effectiveness and applicability of our forecasting approaches. Figures 5 and 6 and Figure 7respectively display partial predictive results of our $\text{S}^2\text{DBM}$ model on the ETTh1, ETTm1, and Weather datasets.

### A.3 EXPERIMENTAL DETAILS

#### A.3.1 DATASET INFORMATION

We adopt seven real-world benchmarks in the experiments to evaluate the accuracy of multivariate time series forecasting, Table 6 summarizes the statistics of these datasets. We adopted the experimental settings from recent studies (Liu et al., 2023b; Zeng et al., 2023; Li et al., 2023b). Specifically, following the recommendations of Dlinear (Zeng et al., 2023), we set the input length $H = 336$. We assessed the prediction accuracy for lengths $L = \{96, 192, 336, 720\}$ across the Weather, Exchange, ETTh1, ETTh2, ETTm1, and ETTm2 datasets, and $L = \{24, 36, 48, 60\}$ for the ILI dataset.

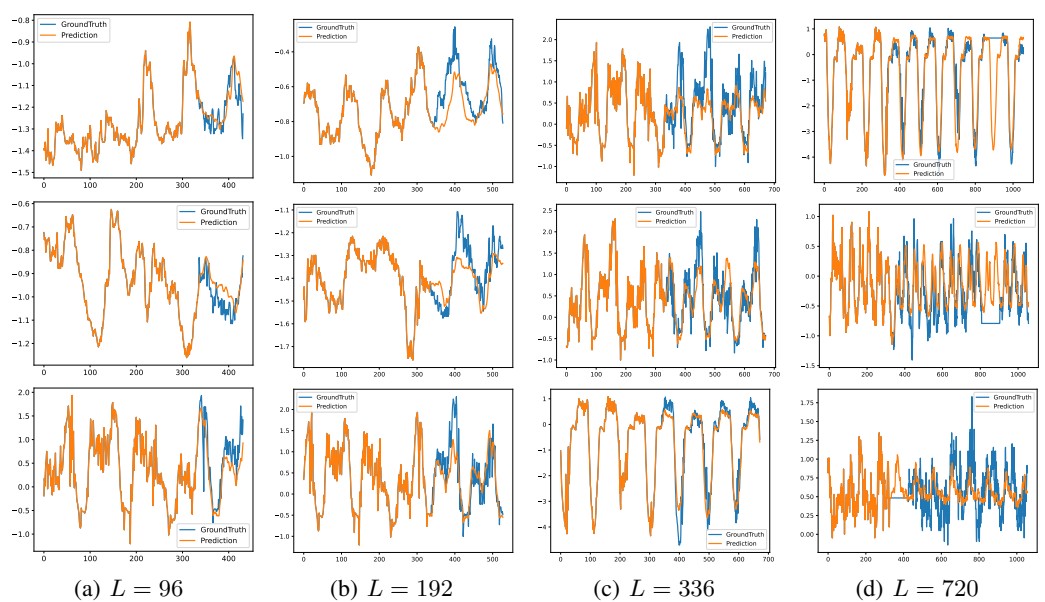

Figure 6: The predicted samples by our $S^2$DBM model for different forecast window lengths on the ETTm1 dataset.

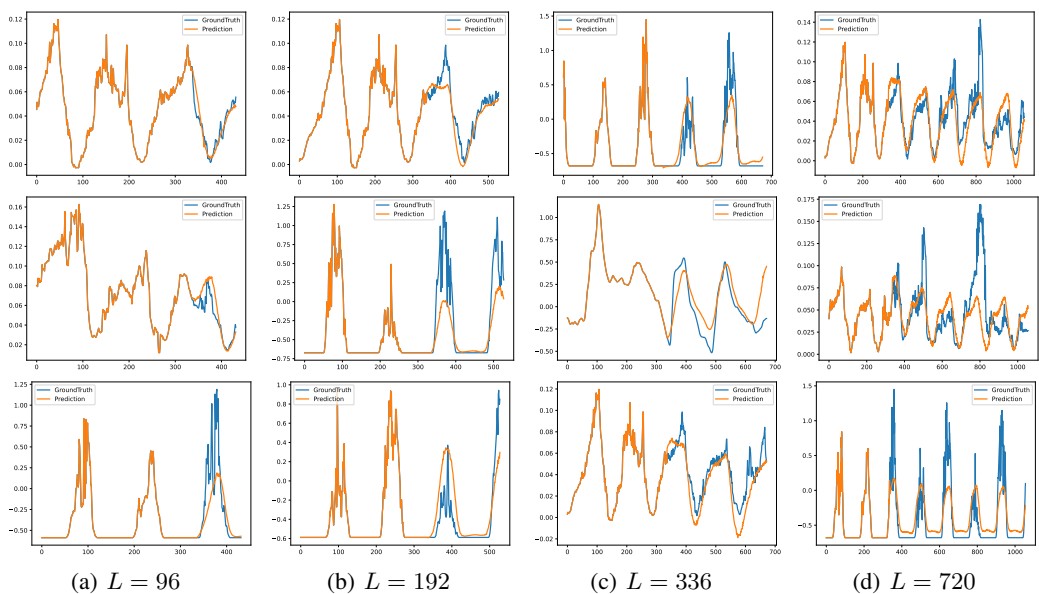

Figure 7: The predicted samples by our $S^2$DBM model for different forecast window lengths on the Weather dataset.

Table 6: Brief statistics of the datasets.

| Datasets | Channels | Granularity | Timesteps |
|---|---|---|---|
| Weather | 21 | 10 min | 59696 |
| ILI | 7 | 1 week | 966 |
| Exchange | 8 | 1 day | 7588 |
| ETTh1&ETTh2 | 7 | 1 hour | 17420 |
| ETTm1&ETTm2 | 7 | 5 min | 69680 |

Table 7: $S^2DBM$ hyperparameters.

| Hyperparameter | Value |
|---|---|
| Residual layers | 4 |
| Residual channels | 8 |
| Diffusion embedding dim | 8 |
| Schedule | Linear |
| Diffusion steps $T$ | 50 |
| Self-attention layers time dim | 1 |
| Self-attention heads time dim | 8 |
| Self-attention layers feature dim | 1 |
| Self-attention layers time dim | 8 |
| EMA decay | 0.995 |
| EMA update interval | 8 |
| Optimizer | Adam |
| Loss function | MAE |
| Max learning rate | $1 \times 10^{-4}$ |
| Min learning rate | $5 \times 10^{-7}$ |
| Individual channels | False |

### A.3.2 IMPLEMENTATION DETAILS

As mentioned in Section 3.3, the denoising network of $S^2DBM$ adopts the same architecture as CSDI Tashiro et al. (2021) but removes modules related to its original conditioning mechanism. Both the conditional encoder network $E$ and the prior predictor $F(\cdot)$ in $S^2DBM$ employ a simple one-layer linear model (Zeng et al., 2023). Table 7 contains the hyperparameters that for $S^2DBM$ training and architecture.

## A.4 ADDITIONAL RESULTS AND EXPERIMENTS

### A.4.1 PROBABILISTIC FORECASTING PERFORMANCE

This section summarizes the probabilistic forecasting results for prediction horizons of 192 and 336, as presented in Table 8 and Table 9. The results demonstrate that our $S^2DBM$ competes effectively with CSDI and TMDM, showcasing competitive performance in terms of CRPS and $\text{CRPS}_{\text{sum}}$ for longer horizon settings.

Table 8: Probabilistic forecasting performance comparisons in terms of CRPS and $\text{CRPS}_{\text{sum}}$. The best results are boldfaced. The prediction horizon set to 192.

| Dataset | ETTh1 | | ETTh2 | | ETTm1 | | ETTm2 | | Weather | |
|---|---|---|---|---|---|---|---|---|---|---|
| Metric | CRPS | $\text{CRPS}_{\text{sum}}$ | CRPS | $\text{CRPS}_{\text{sum}}$ | CRPS | $\text{CRPS}_{\text{sum}}$ | CRPS | $\text{CRPS}_{\text{sum}}$ | CRPS | $\text{CRPS}_{\text{sum}}$ |
| CSDI | 0.544±0.101 | 1.789±0.002 | 1.002±0.126 | 4.827±0.004 | 0.426±0.104 | 1.761±0.001 | 0.465±0.105 | 2.620±0.001 | **0.180±0.024** | **1.604±0.001** |
| TMDM | 0.471±0.087 | **1.729±0.002** | **0.383±0.121** | **1.800±0.003** | 0.369±0.097 | 1.757±0.001 | 0.292±0.105 | **1.375±0.001** | 0.239±0.031 | 1.895±0.001 |
| Ours | **0.406±0.097** | 1.871±0.002 | 0.384±0.102 | 1.816±0.003 | **0.355±0.092** | **1.675±0.001** | **0.288±0.080** | 1.417±0.001 | 0.247±0.031 | 2.171± 0.001 |

Table 9: Probabilistic forecasting performance comparisons in terms of CRPS and $\text{CRPS}_{\text{sum}}$. The best results are boldfaced. The prediction horizon set to 336.

| Dataset | ETTh1 | | ETTh2 | | ETTm1 | | ETTm2 | | Weather | |
|---|---|---|---|---|---|---|---|---|---|---|
| Metric | CRPS | $\text{CRPS}_{\text{sum}}$ | CRPS | $\text{CRPS}_{\text{sum}}$ | CRPS | $\text{CRPS}_{\text{sum}}$ | CRPS | $\text{CRPS}_{\text{sum}}$ | CRPS | $\text{CRPS}_{\text{sum}}$ |
| CSDI | 0.616±0.108 | 2.349±0.002 | 0.928±0.101 | 5.039±0.003 | 0.454±0.095 | 1.808±0.001 | 0.626±0.092 | 2.702±0.001 | 0.358±0.044 | 3.229±0.002 |
| TMDM | 0.524±0.095 | 1.901±0.002 | **0.395±0.099** | **1.769±0.002** | 0.380±0.099 | 1.889±0.001 | 0.464±0.147 | 2.260±0.001 | 0.280±0.035 | 2.307±0.001 |
| Ours | **0.418±0.102** | **1.851±0.002** | 0.422±0.101 | 2.019±0.002 | **0.373±0.095** | **1.764±0.001** | **0.320±0.090** | **1.561± 0.001** | **0.247±0.031** | **2.171±0.001** |

### A.4.2 THE IMPACT OF THE NUMBER OF DIFFUSION STEPS

This section explores the effect of the number of diffusion steps on model performance. Models were trained on the ETTh1 dataset with varying diffusion step counts and evaluated using a prediction length of 96. The results are presented in Table 10. The results indicate strong robustness across different diffusion steps, confirming the model's adaptability to changes in this parameter.

Table 10: The impact of the number of diffusion steps on model performance.

| Model | Diffusion Steps | Training time | Sampling time | MSE | MAE |
|---|---|---|---|---|---|
| $S^2DBM$ | 50 | 88 mins | 852 seconds | 0.3660 | 0.3836 |
| $S^2DBM$ | 200 | 97 mins | 3413 seconds | 0.3659 | 0.3835 |
| $S^2DBM$ | 1000 | 125 mins | 16935 seconds | 0.3656 | 0.3834 |

### A.4.3 THE IMPACT OF THE DIFFERENT CHOICES OF PRIOR PREDICTOR

To validate the impact of different implementations of prior predictor $F(\cdot)$, we conduct an ablation study on the ETTh1 dataset. Specifically, $F(\cdot)$ was varied among a Linear model, NLinear model, DLinear model, and Transformer model for point forecasting with a prediction horizon of 96. The results, summarized in Table 11, highlight consistent performance across these variations, reinforcing our choice of the Linear model for its simplicity, efficiency, and effectiveness.

Table 11: The impact of the different choices of $F(\cdot)$ on model performance and parameter numbers.

| | Linear | NLinear | DLinear | Transformer |
|---|---|---|---|---|
| MSE | 0.366 | 0.335 | 0.366 | 0.365 |
| Num of parameter | 0.05M | 0.05M | 0.10M | 10.54M |

### A.4.4 INFERENCE EFFICIENCY

To offer a clear perspective on the performance of $S^2DBM$, particularly for larger datasets and real-time forecasting applications, we conducted targeted tests on the ETTh1 and Weather datasets. The prediction horizon $L$ was varied to evaluate the inference efficiency of the proposed $S^2DBM$. Table 12 summarizes the inference time for multivariate forecasting with different prediction lengths $L$ on the ETTh1 and Weather datasets.

Table 12: Inference time (ms) on the multivariate forecasting with different prediction horizon L.

| | L=96 | L=192 | L=336 | L=720 |
|---|---|---|---|---|
| ETTh1 | 433.7 | 456.9 | 409.5 | 627.6 |
| Weather | 738.8 | 814.0 | 834.4 | 894.1 |

### A.4.5 ROBUSTNESS TESTING

To evaluate the resilience of our $S^2DBM$ model under adverse conditions with noisy inputs, we introduce noise to the known time series $y$ as follows:

$$\boldsymbol{y}_{\text{noisy}} = \boldsymbol{y} + a \cdot \epsilon, \quad \epsilon \sim \mathcal{N}(\mathbf{0}, \mathbf{I}).$$

The noisy data $y_{\text{noisy}}$ is then used as input for the $\text{S}^2\text{DBM}$ model, and its predictive performance is monitored across various noise levels by adjusting the coefficient $a$. Experimental results in Table 13 indicate that the $\text{S}^2\text{DBM}$ model exhibits robust performance against input noise.

Table 13: The robustness testing on ETTh2 dataset.

| a | L=96 MSE | MAE | L=192 MSE | MAE | L=336 MSE | MAE | L=720 MSE | MAE |
|---|---|---|---|---|---|---|---|---|
| 0 | 0.274 | 0.331 | 0.354 | 0.388 | 0.433 | 0.454 | 0.592 | 0.568 |
| 5% | 0.275 | 0.332 | 0.355 | 0.389 | 0.427 | 0.453 | 0.591 | 0.568 |
| 10% | 0.276 | 0.334 | 0.356 | 0.390 | 0.429 | 0.454 | 0.592 | 0.569 |
| 25% | 0.284 | 0.348 | 0.362 | 0.399 | 0.434 | 0.459 | 0.600 | 0.572 |
| 50% | 0.312 | 0.384 | 0.385 | 0.426 | 0.452 | 0.476 | 0.625 | 0.585 |

