# OpenReview forum: "Series-to-Series Diffusion Bridge Model"
_ICLR.cc/2025/Conference — Submitted to ICLR 2025_

### Official Review · Reviewer_AQwz · 2024-10-28

**Soundness:** 3
**Presentation:** 3
**Contribution:** 3
**Rating:** 8
**Confidence:** 3

**Summary:**

This paper addresses point-to-point time-series forecasting through the use of diffusion models. The authors propose a unified non-autoregressive framework that encompasses most existing diffusion-based time-series models. Building on this framework, they introduce S2DBM, which incorporates the Brownian Bridge process. By integrating historical information via informative priors and conditions, S2DBM effectively reduces randomness and enhances forecasting accuracy.

**Strengths:**

1. The presentation of this paper is clear and easy to follow, with algorithms well articulated.
2. The paper provides a unified framework for several existing diffusion-based time series models, which is well-supported by theoretical foundations. The insights offered are impressive.
3. Both point-to-point and probabilistic forecasting are addressed in the reverse process, broadening the algorithm's application scope.

**Weaknesses:**

1. Some expressions reference prior work without sufficient explanations, which can hinder comprehension. For instance, in the last equation of Section 3.1, the introduction of \(y_\theta\) relies solely on a citation, making it time-consuming for readers to fully grasp its meaning.
2. Table 4 presents results only for a horizon of 96, omitting long horizon settings. This limits the analysis and may leave important insights unaddressed.

**Questions:**

1. In Table 2, your models do not achieve the best performance across all datasets. Could you explain why linear models perform best in several cases, while your models do not?
2. Since $c = E(x)$ and $h = F(x)$ are two important components of your algorithms, I am curious about the role of F in enhancing performance. What impact does it have? An ablation study of F may help.

---

> ### Author Response · Authors · 2024-11-21
>
> **About the introduction of $y_\theta$ :** Thank you for your constructive feedback. We acknowledge that certain references to prior work, including the introduction of  $y_\theta$ in Section 3.1, have been cited without adequate contextual explanation, potentially making it challenging for readers to understand their significance directly from our manuscript. To address this, we have revised the manuscript to include a more detailed explanation of $y_{\theta}$ in Lines 173-184.
>
> **About the probabilistic forecasting performance**  Thank you for your valuable feedback. We have addressed this concern by including additional results for horizons of 192 and 336 in Table 8 and Table 9 of the revised manuscript. The extended results demonstrate that our Series-to-Series Diffusion Bridge Model (\(\mathrm{S^2DBM}\)) competes effectively with CSDI and TMDM, showcasing competitive performance in terms of $\mathrm{CRPS}$ and $\mathrm{CRPS}_{\mathrm{sum}}$ for longer horizon settings.
>
> | Dataset | ETTh1           | ETTh1           | ETTh2           | ETTh2           | ETTm1           | ETTm1           | ETTm2           | ETTm2           | Weather         | Weather         |
> | ------- | --------------- | --------------- | --------------- | --------------- | --------------- | --------------- | --------------- | --------------- | --------------- | --------------- |
> | Metric  | CRPS            | CRPS_sum        | CRPS            | CRPS_sum        | CRPS            | CRPS_sum        | CRPS            | CRPS_sum        | CRPS            | CRPS_sum        |
> | CSDI    | 0.544±0.101     | 1.789±0.002     | 1.002±0.126     | 4.827±0.004     | 0.426±0.104     | 1.761±0.001     | 0.465±0.105     | 2.620±0.001     | **0.180±0.024** | **1.604±0.001** |
> | TMDM    | 0.471±0.087     | **1.729±0.002** | **0.383±0.121** | **1.800±0.003** | 0.369±0.097     | 1.757±0.001     | 0.292±0.105     | **1.375±0.001** | 0.239±0.031     | 1.895±0.001     |
> | Ours    | **0.406±0.097** | 1.871±0.002     | 0.384±0.102     | 1.816±0.003     | **0.355±0.092** | **1.675±0.001** | **0.288±0.080** | 1.417±0.001     | 0.247±0.031     | 2.171±0.001     |
>
> | Dataset | ETTh1           | ETTh1           | ETTh2           | ETTh2           | ETTm1           | ETTm1           | ETTm2           | ETTm2           | Weather         | Weather         |
> | ------- | --------------- | --------------- | --------------- | --------------- | --------------- | --------------- | --------------- | --------------- | --------------- | --------------- |
> | Metric  | CRPS            | CRPS_sum        | CRPS            | CRPS_sum        | CRPS            | CRPS_sum        | CRPS            | CRPS_sum        | CRPS            | CRPS_sum        |
> | CSDI    | 0.616±0.108     | 2.349±0.002     | 0.928±0.101     | 5.039±0.003     | 0.454±0.095     | 1.808±0.001     | 0.626±0.092     | 2.702±0.001     | 0.358±0.044     | 3.229±0.002     |
> | TMDM    | 0.524±0.095     | 1.901±0.002     | **0.395±0.099** | **1.769±0.002** | 0.380±0.099     | 1.889±0.001     | 0.464±0.147     | 2.260±0.001     | 0.280±0.035     | 2.307±0.001     |
> | Ours    | **0.418±0.102** | **1.851±0.002** | 0.422±0.101     | 2.019±0.002     | **0.373±0.095** | **1.764±0.001** | **0.320±0.090** | **1.561±0.001** | **0.247±0.031** | **2.171±0.001** |
>
> **About the performance of linear models :** Thank you for your query. Recent studies highlight that linear models, despite their simplicity, often excel due to their explainability and efficiency, sometimes outperforming more complex architectures in specific scenarios. The superior performance of linear models in certain cases remains an area for further exploration. While our Series-to-Series Diffusion Bridge Model (S2DBM) does not always outperform across all scenarios, it achieves commendable results in the majority of them. The effectiveness of linear models in specific instances also provides valuable insights for future enhancements to our model.

---

> ### Author Response · Authors · 2024-11-21
>
> **About the impact of  F:** Thank you for your interest in the functional role of $F(x)$ in our algorithm. In time series forecasting, the lookback and forecast windows often differ, making it challenging for historical sequences to provide structurally informative priors for future targets, unlike the case in image restoration. Therefore, we cannot construct a direct diffusion bridge between historical time series $x$ and future time series $y$. Instead, we employ the prior predictor $F(\cdot)$ to transform historical time series into a deterministic conditional representation $h$. This representation serves as the endpoint for the diffusion process and offers initial guidance for the reverse process, as detailed in Line 347.
>
> To validate the impact of different implementations of $F(\cdot)$, we conduct  an ablation study on the ETTh1 dataset. We vary $F(\cdot)$ among a Linear model, a NLinear model, a DLinear model and a Transformer model for point forecasting with a prediction horizon of 96. The results demonstrate robust performance across these variations, supporting our decision to employ the simple yet effective Linear model due to its efficiency and effectiveness.
>
> | $F(\cdot)$       | Linear | NLinear | DLinear | Transformer |
> | ---------------- | ------ | ------- | ------- | ----------- |
> | MSE              | 0.366  | 0.335   | 0.366   | 0.365       |
> | Num of parameter | 0.05M  | 0.05M   | 0.10M   | 10.54M      |

---

> > ### Comment · Reviewer_AQwz · 2024-11-25
> > **response**
> >
> > I would like to thank the authors for the rebuttal, which addresses most of my concerns.

---

### Official Review · Reviewer_Caas · 2024-11-03

**Soundness:** 2
**Presentation:** 3
**Contribution:** 2
**Rating:** 6
**Confidence:** 3

**Summary:**

This paper presents the Series-to-Series Diffusion Bridge Model ($S^2DBM$), a promising approach in time series forecasting using diffusion models.
Traditional diffusion models often struggle with deterministic point-to-point predictions.
$S^2DBM$ addresses this by leveraging the Brownian Bridge process to reduce noise and improve accuracy in reverse estimations, effectively capturing temporal dependencies in time series data.
The model incorporates historical data as informative priors, stabilizing diffusion and enhancing point-to-point prediction capabilities.
Experimental results on various datasets show that $S^2DBM$ outperforms existing diffusion-based and other state-of-the-art time series models, demonstrating superior accuracy in both deterministic and probabilistic forecasting tasks.

**Strengths:**

S1.
$S^2DBM$ integrates the Brownian Bridge process into time series forecasting using diffusion models.
By redefining the diffusion framework, the authors introduce a new model and consolidate various non-autoregressive diffusion techniques into a comprehensive framework, elucidating their interrelationships and underlying principles.

S2.
The authors conduct thorough theoretical groundwork.
The empirical evaluations are robust, utilizing diverse real-world datasets to benchmark the model's performance against SOTA methods.

S3.
The draft is well-structured and organized.
The introduction succinctly outlines the problem context and motivates the need for the proposed model.

**Weaknesses:**

W1.
Some assumptions and derivations in this work could be better justified.
For example, the choice of using a Brownian Bridge process warrants a more detailed discussion on why it is preferred over other stochastic processes.
Providing empirical evidence or theoretical reasoning for the choice of this process could strengthen the argument for its effectiveness in reducing randomness in forecasting.

W2.
While the paper presents strong performance metrics, it lacks robustness testing under adverse conditions, such as noisy inputs or missing data.
Future experiments should include scenarios with varying levels of data quality to demonstrate how $S^2DBM$ handles real-world challenges.

W3.
Currently, the experiments focus primarily on a single configuration.
Investigating how different settings of hyperparameters, such as the choice of the prior predictor or conditioning modules, impact performance could provide valuable insights.
For instance, evaluating the effects of varying the number of diffusion steps or using alternative conditioning mechanisms could highlight the robustness of $S^2DBM$ across diverse scenarios.

**Questions:**

Q1.
Can you elaborate on the computational efficiency of $S^2DBM$, particularly in relation to larger datasets or real-time forecasting applications?

Q2.
What specific advantages does the Brownian Bridge process offer over other stochastic processes for time series forecasting in the context of your model?

---

> ### Author Response · Authors · 2024-11-21
>
> **About the advantages for Using a Brownian Bridge :** Thank you for your insightful query. As illustrated in Figure 1, while diffusion-based methods are adequate for probabilistic forecasting, they often fall short in point-to-point prediction accuracy, primarily due to inherent randomness. A significant source of this randomness stems from the prior distribution. When CSDI employs a standard diffusion process, it generates targets using standard Gaussian noise as the prior distribution. Although TMDM enhances the prior to be more informative of the target, the prior distribution in TMDM remains confined to a noisy representation, thus providing limited information about the generation target. A Brownian bridge is a continuous-time stochastic process where the probability distribution during the diffusion process is conditioned on both the starting and ending states. This conditioning effectively 'pins down' the process at both ends, enhancing stability by reducing the influence of noisy inputs. By leveraging the framework proposed in Example 1,  our model shifts from a typical data-to-noise process to a data-to-data process, significantly reducing instability from noisy inputs and enabling the accurate generation of future features based on historical time series.
>
> We also modified the TMDM model according to the Brownian bridge we used, and conducted prediction experiments with a prediction lengths of 96 and 192 on the ETTh1 data set. The experimental results show that our modifications can significantly reduce uncertainty.
>
> |         | Var of predictions (L=96) | Var of predictions (L=192) |
> | ------- | ------------------------- | -------------------------- |
> | TMDM    | 0.165                     | 0.187                      |
> | TMDM+BB | 2.078e-13                 | 4.105e-13                  |
>
> **About the robustness testing of  S$^2$DBM:**  Thank you for highlighting the importance of robustness testing. We have  conducted preliminary tests to evaluate the resilience of our S$^2$DBM model under adverse conditions involving noisy inputs. **Please see Section A.4.5 in the revised manuscript**. Specifically, we introduce  noise to the known time series $y$, i.e.,
> $y_{noisy} = y + a \cdot \epsilon, \quad \epsilon \sim \mathcal{N}(\mathbf{0}, \mathbf{I})$.
> We then use  the noisy data $y_{noisy}$ as input for the S$^2$DBM model and monitor  the predictive performance across various noise levels by altering the coefficient \( a \). The experimental results indicate that our S$^2$DBM model demonstrates commendable robustness to noise in inputs.
>
> We acknowledge the need for further comprehensive testing, including scenarios with missing data and varying data quality levels, to more fully ascertain the model's performance under real-world conditions. Future revisions of this work will incorporate these additional robustness tests to provide a clearer demonstration of how S$^2$DBM handles such challenges.
>
> |      | L=96          | L=192         | L=336         | L=720         |
> | ---- | ------------- | ------------- | ------------- | ------------- |
> | a    | MSE / MAE     | MSE / MAE     | MSE / MAE     | MSE / MAE     |
> | 0    | 0.274 / 0.331 | 0.354 / 0.388 | 0.433 / 0.454 | 0.592 / 0.568 |
> | 5%   | 0.275 / 0.332 | 0.355 / 0.389 | 0.427 / 0.453 | 0.591 / 0.568 |
> | 10%  | 0.276 / 0.334 | 0.356 / 0.390 | 0.429 / 0.454 | 0.592 / 0.569 |
> | 25%  | 0.284 / 0.348 | 0.362 / 0.399 | 0.434 / 0.459 | 0.600 / 0.572 |
> | 50%  | 0.312 / 0.384 | 0.385 / 0.426 | 0.452 / 0.476 | 0.625 / 0.585 |

---

> ### Author Response · Authors · 2024-11-21
>
> **About the configuration Settings of  S$^2$DBM:** Thank you for your suggestions on exploring various hyperparameter settings to enhance the robustness and understanding of our S$^2$DBM model. We have undertaken additional experiments to evaluate the impact of different configurations, particularly focusing on the number of diffusion steps (T) and the choice of the prior predictor $F(\cdot)$. Please see section A.4.2 and section A.4.3  in the revised manuscript.
>
> *(1) Diffusion Steps Impact:*  We experiment  with varying the number of diffusion steps in the S$^2$DBM model. Our results indicate strong robustness across different diffusion steps, confirming the model's adaptability to changes in this parameter.
>
> | Diffusion Steps | Training time | Sampling time | MSE    | MAE    |
> | --------------- | ------------- | ------------- | ------ | ------ |
> | 50              | 88 mins       | 852s          | 0.3660 | 0.3836 |
> | 200             | 97 mins       | 3413s         | 0.3659 | 0.3835 |
>
> *(2) About the ablation Study on $F(\cdot)$:* Additionally, to validate the impact of different implementations of $F(\cdot)$, we conduct  an ablation study on the ETTh1 dataset. We varied $F(\cdot)$ among   a Linear model adopted in S$^2$DBM, a NLinear model , a DLinear model, and a non-linear Transformer model for point forecasting with a prediction horizon of 96. The results demonstrate robust performance across these variations, supporting our decision to employ the simple yet effective Linear model due to its efficiency and effectiveness.
>
> | $F(\cdot)$       | Linear | NLinear | DLinear | Transformer |
> | ---------------- | ------ | ------- | ------- | ----------- |
> | MSE              | 0.366  | 0.365   | 0.366   | 0.365       |
> | Num of parameter | 0.05M  | 0.05M   | 0.10M   | 10.54M      |
>
> **About the computational efficiency of S$^2$DBM:** Thank you for your question regarding the computational efficiency of our  S$^2$DBM  model. To provide a clear perspective on its performance, especially concerning larger datasets and real-time forecasting applications, we have conducted specific tests on the ETTh1 and Weather datasets, varying the prediction horizon \( L \) to assess the inference efficiency of the proposed S$^2$DBM.
>
> |         | L=96   | L=192  | L=336  | L=720  |
> | ------- | ------ | ------ | ------ | ------ |
> | ETTh1   | 433.7s | 456.9s | 409.5s | 627.6s |
> | Weather | 738.8s | 814.0s | 834.4s | 894.1s |
> | 1000            | 125 mins      | 16935s        | 0.3656 | 0.3834 |

---

> ### Author Response · Authors · 2024-11-25
>
> We sincerely thank you once again for your valuable comments. In our rebuttal, we have revised our paper and provided additional results accord to your comments. As the deadline for the public discussion phase approaches, we would greatly appreciate your response to our rebuttal. We are ready and willing to address any additional concerns you may have.

---

> > ### Comment · Reviewer_Caas · 2024-11-27
> > **Thanks for the authors' rebuttals**
> >
> > Thanks to the authors' efforts in responding to my concerns.
> > Most of my concerns are addressed in the revised manuscript.
> > I will consider raising my ratings accordingly.

---

### Official Review · Reviewer_ZMdS · 2024-11-04

**Soundness:** 2
**Presentation:** 2
**Contribution:** 2
**Rating:** 6
**Confidence:** 2

**Summary:**

This paper revisits the application of diffusion models for time series forecasting, presenting a unified framework that consolidates these methods. Building on this framework, the authors incorporate the Brownian bridge process to enhance prediction accuracy. The results demonstrate that the proposed approach outperforms existing diffusion-based forecasting models.

**Strengths:**

The authors provide a comprehensive summary of existing models, highlighting that their primary differences lie in the formulation of $\hat{\gamma_t}$. By introducing the Brownian bridge process into diffusion-based time series forecasting models, they establish a range of relevant properties.

**Weaknesses:**

1. In Line 259, the title of Proposition 1 is "Brownian Bridge between Historical and Predicted Time Series." However, the Brownian bridge’s endpoint is set to $h$, the Prior Predictor’s forecasted value of $x$. The paper provides no explanation as to why $h$ is chosen as the endpoint for the Brownian bridge.
2. It would be beneficial for the authors to clarify, from a theoretical perspective, why the Brownian bridge is integrated into the diffusion model for time series forecasting. Specifically, how does its ability to "pin down" the diffusion process at both ends help reduce instability from noisy inputs and enable the accurate generation of future features based on historical time series?
3. In line 195, Theorem 1 concerns non-autoregressive diffusion processes. However, the summarized models, CSDI and SSSD, do not appear to be non-autoregressive models. Could there be an issue with the theorem here?

**Questions:**

See weakness plz.

---

> ### Author Response · Authors · 2024-11-21
>
> **About  why $h$ is chosen as the endpoint for the Brownian bridge:**  In time series forecasting, it is common for the lengths of the lookback and forecast windows to differ, while diffusion bridges require the lengths at both ends to be the same. Therefore, we cannot directly construct a diffusion bridge between historical time series $x$ and future time series $y$. Instead, we use the prior predictor $F(\cdot)$ to transform historical time series into a deterministic conditional representation $h$, which serves as the endpoint of the diffusion process and provides guidance at the beginning of the reverse process.  We have provided explanation in the original manuscriput and it is in Line 347 of the revised version.
>
> **About the Theoretical Justification for Integrating Brownian Bridge into Diffusion Models for Time Series Forecasting :** Thank you for your insightful query. As illustrated in Figure 1, while diffusion-based methods are adequate for probabilistic forecasting, they often fall short in point-to-point prediction accuracy, primarily due to inherent randomness. A significant source of this randomness stems from the prior distribution. When CSDI employs a standard diffusion process, it generates targets using standard Gaussian noise as the prior distribution. Although TMDM enhances the prior to be more informative of the target, the prior distribution in TMDM remains confined to a noisy representation, thus providing limited information about the generation target.
> A Brownian bridge is a continuous-time stochastic process where the probability distribution during the diffusion process is conditioned on both the starting and ending states. This conditioning effectively 'pins down' the process at both ends, enhancing stability by reducing the influence of noisy inputs [1-4]. By leveraging a Brownian bridge, our model shifts from a typical data-to-noise process to a data-to-data process, significantly reducing instability from noisy inputs and enabling the accurate generation of future features based on historical time series.
>
> [1] Yue C, Peng Z, Ma J, et al. Image restoration through generalized ornstein-uhlenbeck bridge.
>
> [2] Liu G H, Vahdat A, Huang D A, et al. I $^ 2$ SB: Image-to-Image Schr\" odinger Bridge.
>
> [3] Wang Y, Yoon S, Jin P, et al. Implicit Image-to-Image Schrödinger Bridge for Image Restoration.
>
> [4] Chen Z, He G, Zheng K, et al. Schrodinger bridges beat diffusion models on text-to-speech synthesis.
>
> **About the determination of Non-Autoregressive Diffusion Processes:** As far as we are aware, CSDI and SSSD are not autoregressive models. Autoregressive time-series diffusion models generate future predictions sequentially over time. In contrast, CSDI and SSSD avoid autoregressive inference by simultaneously diffusing and denoising the entire time series, making them non-autoregressive diffusion models. A more detailed discussion of the categorization of time-series diffusion models can be found in [5], which explicitly demonstrates that CSDI and SSSD are not autoregressive models.
>
> [5] Shen, Lifeng, and James Kwok. "Non-autoregressive conditional diffusion models for time series prediction." International Conference on Machine Learning. PMLR, 2023.

---

### Official Review · Reviewer_fQ7V · 2024-11-11

**Soundness:** 2
**Presentation:** 3
**Contribution:** 2
**Rating:** 3
**Confidence:** 4

**Summary:**

This paper presents a comprehensive framework that encompasses most existing diffusion-based methods. Building on this foundation, the authors introduce the Series-to-Series Diffusion Bridge Model (S2DBM). Experimental results demonstrate that S2DBM delivers superior performance.

**Strengths:**

This paper utilizes the Diffusion Bridge Model to help the reverse process start from a more deterministic state, reducing the instability caused by noise and thereby facilitating better predictions.

**Weaknesses:**

weakness:
1. There is a notation issue with \(\hat{\gamma}_t\) in line 203; the writing needs to be standardized. Additionally, it needs to be clarified whether the values of \(\hat{\alpha}_t\), \(\hat{\beta}_t\), and \(\gamma_t\) should have a specific relationship to conform to the diffusion model.
2. In line 411, it is mentioned that a comparison with timediff was made, but Table 2 does not include TimeDiff data while other baselines are present.
3. The content of the paper appears to primarily build on existing work by combining the model guidance from TMDM and the non-autoregressive approach of timediff with the existing Brownian Bridge process. This integration seems to lack novelty.

**Questions:**

NA

---

> ### Author Response · Authors · 2024-11-21
>
> **About the  parameters of  forward process:**  We appreciate your attention to the typo regarding the inconsistencies, which has been corrected in line 206.  As outlined in Theorem 1, Eq. (1) provides a general framework for the forward process. The proposed framework only requires that the parameters be designed to ensure   that $x_t$ remains pristine at $t=0$ and undergoes maximal degradation at $t=T$. We have clarified this point in the revised version;  please see line 206 in the revised manuscript. Moreover, as demonstrated in Table 1, $\hat{\alpha}_t$, $\hat{\beta}_t$, and $\hat{\gamma}_t$ offer significant flexibility, enabling various parameterization techniques that confer distinct statistical characteristics upon the diffusion models.  The specific values for $\hat{\alpha}_t$, $\hat{\beta}_t$, and $\hat\gamma_t$ in our proposed S$^2$DBM are detailed in Eq. (4) (line 267), Example 1 (line 310), and Example 2 (line 319).
>
> **About  the comparison with TimeDiff :** We appreciate the reviewer’s feedback and would like to clarify the rationale behind the organization of our results.  We kindly note that the comparison with TimeDiff is presented in  *Table 3* of the *original manuscript*. The comparison with TimeDiff is not included in Table 2 because TimeDiff uses unique settings for prediction length, which differs from those of most methods in Table 2.  While TimeDiff's official code has not been publicly released, we have diligently retrained our model following the settings described in the TimeDiff paper. Subsequently, we included a performance comparison with the results reported in the TimeDiff paper in Table 3 to ensure a *fair* and accurate evaluation. We hope this explanation addresses the reviewer’s concerns and provides clarity regarding the alignment of our experimental setup with the existing literature.
>
> **About the novelty of our work:** We would like to clarify that our work goes beyond a simple combination of TMDM and TimeDiff.  The contribution of our approach lies in reducing the instability caused by noisy input and enabling the accurate generation of future time-step features from historical time series.   The prior distributions in both TMDM and TimeDiff are constrained to noisy representations, providing limited guidance for generating the target outputs. In contrast, our approach introduces a Brownian Bridge process, which operates as a data-to-data diffusion mechanism rather than the conventional data-to-noise process used in existing diffusion models. This *paradigm shift* mitigates the instability caused by noisy input and facilitates the accurate generation of future time-step features based on historical time series data.
>
> Moreover, we contribute a novel and unified framework that integrates and generalizes existing non-autoregressive diffusion-based time series forecasting models, including TMDM and TimeDiff. This theoretical insight offers a new perspective on the design of diffusion models for time series prediction, making our contribution conceptually significant. To substantiate the distinctiveness and effectiveness of our proposed framework, we conducted additional experiments by incorporating TMDM  (whose code is off the shelf) into our framework. Notably, we preserve  TMDM’s original network architecture and modify only the forward and reverse processes as outlined in  Example 1  of our paper. We evaluate this  model on the ETTh2 dataset with a prediction length of 192. The experimental results, summarized below, clearly demonstrate that our framework significantly enhances TMDM’s performance:
>
> | Model   | MSE   | MAE   |
> | ------- | ----- | ----- |
> | TMDM    | 0.578 | 0.535 |
> | TMDM+Our framework | 0.528 | 0.471 |

---

> > ### Author Response · Authors · 2024-11-25
> >
> > We sincerely thank you once again for your valuable comments. As the deadline for the public discussion phase approaches, we would greatly appreciate your response to our rebuttal. We are ready and willing to address any additional concerns you may have.

---

> > > ### Author Response · Authors · 2024-12-01
> > >
> > > To further highlight the distinct contributions of the proposed framework, we integrate the TMDM model (available as off-the-shelf code) into our framework and conduct prediction experiments with horizons of 96 and 192 on the ETTh1, ETTh2, ETTm1, and ETTm2 datasets.  The experimental results demonstrate that our framework significantly improves the accuracy of TMDM while enhancing stability by reducing the influence of noisy inputs compared to TMDM alone.
> > >
> > > |    Dataset    |       ETTh1       |      ETTh2       |       ETTm1       |       ETTm2       |
> > > | :-----------: | :---------------: | :--------------: | :---------------: | :---------------: |
> > > | Metric (L=96) |     MSE / MAE     |    MSE / MAE     |     MSE / MAE     |     MSE / MAE     |
> > > |     TMDM      |   0.711 / 0.605   |  0.496 / 0.510   |   0.547 / 0.512   |   0.328 / 0.400   |
> > > |    TMDM+BB    | **0.495 / 0.457** | **0.377 /0.396** | **0.426 / 0.418** | **0.235 / 0.303** |
> > >
> > >
> > > |    Dataset     |       ETTh1       |       ETTh2       |       ETTm1       |       ETTm2       |
> > > | :------------: | :---------------: | :---------------: | :---------------: | :---------------: |
> > > | Metric (L=192) |     MSE / MAE     |     MSE / MAE     |     MSE / MAE     |     MSE / MAE     |
> > > |      TMDM      |   0.922 / 0.720   |   0.578 / 0.535   |   0.689 / 0.488   |   0.415 / 0.423   |
> > > |    TMDM+BB     | **0.553 / 0.491** | **0.522 / 0.454** | **0.488 / 0.460** | **0.392 / 0.356** |
> > >
> > > | ETTh1   | Var of predictions (L=96) | Var of predictions (L=192) |
> > > | ------- | ------------------------- | -------------------------- |
> > > | TMDM    | 0.165                     | 0.187                      |
> > > | TMDM+BB | 2.078e-13                 | 4.105e-13                  |
> > >
> > > | ETTh2   | Var of predictions (L=96) | Var of predictions (L=192) |
> > > | ------- | ------------------------- | -------------------------- |
> > > | TMDM    | 0.066                     | 0.078                      |
> > > | TMDM+BB | 1.577e-12                 | 1.525e-12                  |
> > >
> > > | ETTm1   | Var of predictions (L=96) | Var of predictions (L=192) |
> > > | ------- | ------------------------- | -------------------------- |
> > > | TMDM    | 0.161                     | 0.125                      |
> > > | TMDM+BB | 5.003e-13                 | 5.047e-13                  |
> > >
> > > | ETTm2   | Var of predictions (L=96) | Var of predictions (L=192) |
> > > | ------- | ------------------------- | -------------------------- |
> > > | TMDM    | 0.054                     | 0.064                      |
> > > | TMDM+BB | 1.546e-12                 | 1.639e-12                  |
> > >
> > > Thank you for your detailed comments and constructive suggestions on our paper. We have carefully addressed your concerns in our rebuttal and provided additional experimental results to clarify the distinct contributions of our framework.
> > >
> > > As the discussion phase is coming to a close, we would greatly appreciate any further feedback or follow-up questions you might have regarding our responses or the additional analyses.
> > >
> > > Thank you again for your time and thoughtful review!

---

### Meta-Review · Area_Chair_xWBx · 2024-12-23

**Metareview:**

The paper introduces a unified framework for diffusion models for time series forecasting and, building on this framework, it introduces the Brownian bridge process to enhance prediction accuracy. In terms of overall ratings, one reviewer is positive, two reviewers are mildly positive and one reviewer is negative. However, the two mildly positive reviews do appear lukewarm -- one reviewer increased their score during the rebuttal period as an "encouragement" -- and they do share some of the same concerns raised by the reviewer with an overall negative rating. The main issues comprise the significance of the contribution, the justification/motivation to use the Brownian bridge, and the experimental evaluation. As a consequence, I am unable to recommend acceptance.

**Additional Comments On Reviewer Discussion:**

The authors put a significant effort in addressing the reviewers' concerns during the rebuttal period and two of the reviewers increased their score. However, as argued in the metareview, the reviews themselves appear lukewarm.

---

### Decision · Program_Chairs · 2025-01-22

Reject